# DATA SELECTION FOR LLM REINFORCEMENT LEARNING WITH IMPROVED GRADIENT ALIGNMENT

## ABSTRACT

Reinforcement Learning with Verifiable Rewards (RLVR) has become a key technique for enhancing LLMs' reasoning abilities, yet its data inefficiency remains a major bottleneck. To address this critical yet challenging issue, we present a novel gradient-alignment-based method, named *LearnAlign*, which intelligently selects the learnable and representative training reasoning data for RLVR post-training. To overcome the issue of response-length bias in gradient norms, we introduce the data learnability based on the success rate, which can indicate the learning potential of each data point. Experiments across five reasoning benchmarks show that our method significantly reduces training data requirements while achieving minor performance degradation or even improving performance compared to full-data training. Specifically, it reduces data requirements by up to 1,000 data points with better performance (77.5%) than that on the full dataset on the GSM8K benchmark (77.0%). Furthermore, its efficiency is demonstrated on both mathematical and code benchmarks by using much less data from the DAPO-MATH-17K dataset. We believe this work provides some insights for data-efficient RL post-training and could help future research on reasoning data selection. To facilitate future work, we will release code.

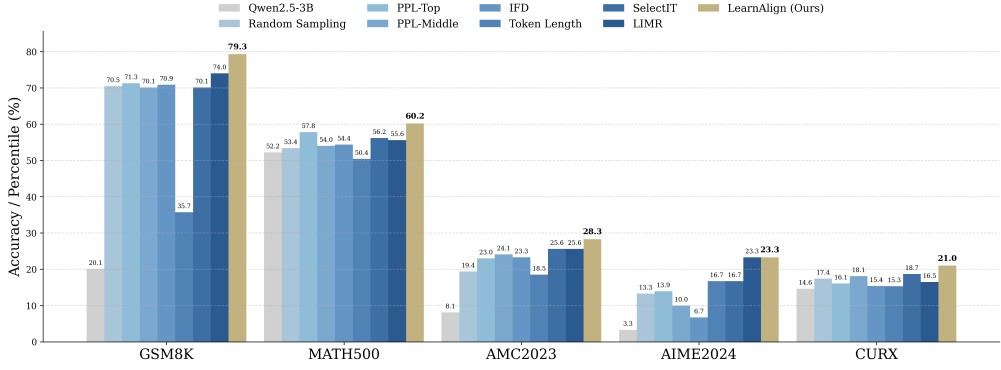

Figure 1: Performance comparison between baseline methods and our proposed *LearnAlign* on various benchmarks, including GSM8K, MATH500, AMC2023, AIME2024, and CRUX, using the Qwen2.5-3B model.

## 1 INTRODUCTION

Recently, Reinforcement Learning (RL) has become a successful and crucial post-training paradigm for enhancing the reasoning ability of large language models (LLMs), exemplified by OpenAI o1 (Jaech et al., 2024), DeepSeek-R1 (Guo et al., 2025), Kimi k1.5 (Team et al., 2025), and so on. These models commonly utilize a rule-based reward function, such as the correctness of mathematical solving and code generation problems, to provide the supervision signal.

Due to the large number of parameters, the post-training for LLMs usually needs a lot of computing resources with large-scale data (Zhou et al., 2023; Luo et al., 2024; 2025; Liu et al., 2025a; Li et al.,

2022; Zhang et al., 2024). While, according to the recent studies (Zhou et al., 2023; Ye et al., 2025), it is feasible to activate the specialized ability of a pre-trained language model in downstream tasks with a small set of examples. Inspired by this observation, several works (Xia et al., 2024a; Li et al., 2023a; Liu et al., 2024a) have explored data selection strategies for the post-training of LLMs. Most of these data selection methods obtain a quality score for each data point by utilizing an external expert model or the learning signals of the model that needs to be trained, and then select the top-sorted data with scores. While, these works are specially designed for the supervised fine-tuning paradigm rather than the reinforcement learning paradigm, which shows limited effectiveness in reasoning-oriented scenarios. As far as we know, there are very few works (Li et al., 2025; Wang et al., 2025) that studied the data selection problem of the reinforcement learning paradigm at present. These works (Li et al., 2025; Wang et al., 2025) verified that a small amount of data or even one training example can still provide sufficient information for RLVR post-training. However, their methods are not computationally efficient, since they need to train the whole original dataset for several epochs during data selection, which makes them less practical for saving computing resources.

To address the above issue, we propose a practical data selection method, named *LearnAlign*, for the RLVR paradigm in large language models via gradient alignment. Inspired by (Pruthi et al., 2020; Xia et al., 2024a), to select the high-valued reasoning data, we consider measuring the influence of each data point for training the LLM. First, we estimate the influence of one data point for the training dataset by approximating the change of the training loss using a first-order Taylor expansion. Such influence then can be transformed to the alignment score of gradients between that data point and the training dataset, which can reflect the representativeness of data points to the dataset. In addition, to address the well-known response-length bias for gradient norms (Liu et al., 2025b; Xia et al., 2024a), we introduce the learning ability of data estimated by the success rate to replace it, which can represent the learnable potential without the bias (Florensa et al., 2018; Tzannetos et al., 2023). Finally, we can obtain an improved gradient alignment score, and then the top-sorted data points are identified as the learnable and representative reasoning data.

Experiments across four mathematical reasoning benchmarks (GSM8K (Cobbe et al., 2021), MATH500 (Hendrycks et al., 2021a), AMC2023 (AMC, 2023), and AIME2024 (AIM, 2024)) and one code generation benchmark (CRUX (Gu et al., 2024)) reveal two key findings: (1) conventional SFT data selection methods fall short in the RLVR paradigm for the post-training phase of LLMs; (2) *LearnAlign* achieves minor performance degradation or even superior performance while requiring only a fraction of the training data (As seen in Figure 1). Notably, our method achieves comparable performance compared to full data (42.4% vs. 44.9%) using much less data (1,000 vs. 17,000 examples) across five benchmarks.

Our main contributions are summarized as follows:

- In this paper, we explore efficient data selection for RLVR post-training from the perspective of gradient alignment, a direction that has received limited attention in prior work.
- We introduce *LearnAlign*, a novel data selection framework that constructs learnability-weighted gradient representations to measure influence between data points, where the learnability metric captures learning potential and addresses the response-length bias for gradient norms.
- Comprehensive comparison with prior methods across five benchmarks and three LLMs clearly reveals the shortcomings of traditional SFT data selection methods, and demonstrates that LearnAlign identifies high-value subsets that match or exceed full-dataset performance.

## 2  RELATED WORK

We review the existing data selection studies for LLM post-training, including Supervised Fine-Tuning (SFT) and Reinforcement Learning with Verifiable Rewards (RLVR).

**Data Selection for SFT Post-training:** Commonly, the data selection methods for LLM SFT obtain a quality score for each data point based on different signals. According to the kinds of signals, we can divide these methods into two categories: external-scoring methods and self-scoring methods. For the first category, several recent studies have utilized external LLMs for SFT data scoring and selection. INSTAG (Lu et al., 2023) proposed an open-set instruction tagging framework that employs ChatGPT to generate fine-grained tags, enabling the assessment of instruction diversity and complexity in SFT. Similarly, ALPAGASUS (Chen et al., 2023) leverages ChatGPT to evaluate instruction quality

and selects high-scoring samples for training. IFD (Li et al., 2023a) identifies relevant instruction pairs using a metric measuring discrepancies between model predictions and self-generated outputs. LESS (Xia et al., 2024a) designed a gradient-based selection method that prioritizes data resembling few-shot examples for specific tasks. SelectIT(Liu et al., 2024a) leveraged model uncertainty at multiple levels (token, sentence, and model) to identify high-quality instructions without external supervision. Nuggets (Li et al., 2023b) scores candidate examples by their influence on anchor set perplexity, optimizing instruction tuning efficiency.

**Data Selection for RLVR Post-training:**    As far as we know, there are few works that have explored data selection for RLVR post-training. LIMR (Li et al., 2025) and 1-shot RLVR (Wang et al., 2025) verify earlier that a small amount of data can still provide sufficient information for the scaling of RL. While these methods are not computationally efficient, since they need to train the original dataset for several epochs during data selection. To address this issue, this work offers a more practical solution for RL post-training.

## 3 PRELIMINARY

A next-token prediction LLM can be regarded as a token-level Markov Decision Process (MDP) (Sutton et al., 1998; Foster & Foerster, 2025), which is denoted by a tuple $\mathcal{M} := \{\mathcal{S}, \mathcal{A}, \gamma, \mathcal{T}, \mathcal{R}, \mathcal{P}^0\}$. $\mathcal{S}$ represents the state space, and $\mathcal{A}$ denotes the action space. $\mathcal{P}^0$ means the starting state distribution while $\mathcal{T}$ is the transition function. The reward function and the discount factor are denoted $\mathcal{R}$ and $\gamma$, respectively. LLM post-training by RL is formulated as a token-level MDP, where the objective is to sequentially generate text conditioned on the given prompt. It starts from a prompt or question query denoted as $\boldsymbol{\xi} = [\xi_1, \xi_2, \cdots, \xi_n]$, represents $n$ tokens. At each timestep $t$, the action $y_t \in \mathcal{A}$ corresponds to the generation of a token $y_t$, sampled from the model's output distribution. The transition function $\mathcal{T}([\boldsymbol{\xi}_{0:t-1}, y_t]) = \boldsymbol{\xi}_{0:t}$ is deterministic. It concatenates the generated token $y_t$ to the existing sequence $\boldsymbol{\xi}_{0:t-1} = [\xi_1, \ldots, \xi_n, y_1, \ldots, y_{t-1}]$ to form the new state $\boldsymbol{\xi}_{0:t} = [\xi_1, \ldots, \xi_n, y_1, \ldots, y_t]$. The reward for generating token $y_t$ at timestep $t$ is sparse, assigned only at the final timestep $T$ of the episode. The reward is binary, with $\mathcal{R}(\boldsymbol{\xi}\mathbf{y}) = 1$ if the complete sequence $\boldsymbol{\xi}\mathbf{y} = [\xi_1, \ldots, \xi_n, y_1, \ldots, y_T]$ (the prompt followed by the generated tokens) is correct, and $\mathcal{R}(\boldsymbol{\xi}\mathbf{y}) = 0$ otherwise. Typically, the discount factor $\gamma$ is set to 1, so the cumulative discounted finite-horizon return is simply $\mathcal{R}(\boldsymbol{\xi}\mathbf{y})$.

**Group Relative Policy Optimization (GRPO).**    Recently, GRPO (Shao et al., 2024) emerges as a popular RL algorithm. In this paper, we use it for our experiments. In particular, the GRPO consists of two terms, a policy term $\mathcal{J}_{\text{Policy}}$ and another KL divergence term to constrain the divergence between the old and new policy model. This can be formulated as follows:

$$\mathcal{J}_{\text{GRPO}}(\boldsymbol{\theta}) = \mathbb{E}_{(q,a)\sim\mathcal{P}_q,\{o_i\}_{i=1}^G\sim\pi_{\boldsymbol{\theta}_{old}}(o|q)}$$
$$\left\{\frac{1}{G}\sum_{i=1}^G \frac{1}{|o_i|}\sum_{t=1}^{|o_i|}\min\left[r_{i,t}\hat{A}_{i,t}, \text{clip}(r_t, 1-\varepsilon, 1+\varepsilon)\hat{A}_{i,t}\right] - \beta\mathbb{D}_{\text{KL}}[\pi_{\boldsymbol{\theta}}|\pi_{\text{ref}}]\right\}, \quad (1)$$

where $r_{i,t} = \frac{\pi_{\boldsymbol{\theta}}(o_{i,t}|q,o_{i,<t})}{\pi_{\boldsymbol{\theta}_{\text{old}}}(o_{i,t}|q,o_{i,<t})}$, and $\hat{A}_{i,t}$ denotes the relative advantage, which is computed using a group of rewards $\{r_1, r_2, \cdots, r_G\}$: $\hat{A}_{i,t} = \frac{r_i - \text{mean}(\{r_i\}_{i=1}^G)}{\text{std}(\{r_i\}_{i=1}^G)}$. $\mathbb{D}_{\text{KL}}$ denotes the KL-divergence between $\pi_{\boldsymbol{\theta}}$ and $\pi_{\text{ref}}$. The hyperparameters $\epsilon$ and $\beta$ require tuning, while $\pi_{\text{ref}}$ typically represents the original pre-trained model prior to the RL post-training process.

## 4 METHOD

Here, we outline our strategy for selecting data to effectively enhance the large language model's performance during the reinforcement learning (RL) post-training phase. We begin by defining the data selection problem (Section 4.1). Next, we discuss data influence estimation via gradient alignment (Section 4.2) and improving gradient alignment with data learnability (Section 4.3), which provides a way to assess the utility of data pairs. Finally, we present a comprehensive overview of our data selection method (Section 4.4).

## 4.1 PROBLEM DEFINITION

The objective of data selection for LLM RL post-training is to identify a subset $\mathcal{D}_{\text{train}}^s \subset \mathcal{D}_{\text{train}}$, where $|\mathcal{D}_{\text{train}}^s| < |\mathcal{D}_{\text{train}}|$, from the full training dataset $\mathcal{D}_{\text{train}}$. The selected subset is used to train an LLM policy model $\pi_{\boldsymbol{\theta}}$ via reinforcement learning techniques, e.g., PPO (Schulman et al., 2017) or GRPO (Shao et al., 2024), aiming to achieve lower loss and improved performance on a test dataset $\mathcal{D}_{\text{test}}$. Moreover, no additional information beyond the original training dataset $\mathcal{D}_{\text{train}}$ is available. Ideally, the selected subset should enable the model to achieve performance comparable to training on the full dataset $\mathcal{D}_{\text{train}}$ with significantly fewer data, or ensure that any performance degradation is minimal, thereby maximizing training efficiency.

## 4.2 DATA INFLUENTIAL ESTIMATION VIA GRADIENT ALIGNMENT

Similar to SFT data selection methods (Xia et al., 2024a), selecting data for LLM post-training also requires analyzing and understanding the training dynamics of the data. Specifically, we need to identify which data can most effectively reduce the model's loss. Drawing inspiration from (Pruthi et al., 2020; Liu et al., 2024b), the change in the loss function $\mathcal{J}(\cdot)$ for a given data $\boldsymbol{\xi}$ as the model parameters change from $\boldsymbol{\theta}^t$ to $\boldsymbol{\theta}^{t+1}$ can be approximated using a first-order Taylor expansion as follows:

$$\mathcal{J}(\boldsymbol{\theta}^{t+1}; \boldsymbol{\xi}') \approx \mathcal{J}(\boldsymbol{\theta}^t; \boldsymbol{\xi}') + \nabla\mathcal{J}(\boldsymbol{\theta}^t, \boldsymbol{\xi}')(\boldsymbol{\theta}^{t+1} - \boldsymbol{\theta}^t) + \mathcal{O}(\|\boldsymbol{\theta}^{t+1} - \boldsymbol{\theta}^t\|^2). \quad (2)$$

If the model $\boldsymbol{\theta}^{t+1}$ is trained by a single data $\boldsymbol{\xi}$ with stochastic gradient descent (SGD) at time $t$, this can be expressed as $\boldsymbol{\theta}^{t+1} = \boldsymbol{\theta}^t - \eta_t \nabla\mathcal{J}(\boldsymbol{\theta}^t; \boldsymbol{\xi})$, where $\eta_t$ denotes the learning rate for the time $t$. Substituting this update into Eq.(2), a data $\boldsymbol{\xi}$ update to the model introduces the change of the loss on another sample $\boldsymbol{\xi}'$, which can be formulated as:

$$\begin{aligned}\mathcal{J}(\boldsymbol{\theta}^{t+1}; \boldsymbol{\xi}') - \mathcal{J}(\boldsymbol{\theta}^t; \boldsymbol{\xi}') &\approx \nabla\mathcal{J}(\boldsymbol{\theta}^t; \boldsymbol{\xi}')(\boldsymbol{\theta}^{t+1} - \boldsymbol{\theta}^t) \\ &= -\eta_t \left( \nabla\mathcal{J}(\boldsymbol{\theta}^t; \boldsymbol{\xi}') \cdot \nabla\mathcal{J}(\boldsymbol{\theta}^t; \boldsymbol{\xi}) \right),\end{aligned} \quad (3)$$

where we ignore the higher-order term $\mathcal{O}(\|\boldsymbol{\theta}^{t+1} - \boldsymbol{\theta}^t\|^2)$ as it is small for a sufficiently small step size $\eta_t$. Based on this, we can formalize the influence between two data $\boldsymbol{\xi}_i$ and $\boldsymbol{\xi}_j$.

**Definition 4.1** (Data Influence via Gradient Alignment). *Let $\boldsymbol{\xi}_i$ and $\boldsymbol{\xi}_j$ be two data from the training dataset $\mathcal{D}_{train}$, and let $\boldsymbol{\theta}$ represent the model parameters. The influence of data $\boldsymbol{\xi}_i$ on data $\boldsymbol{\xi}_j$, denoted as $Inf_t(\boldsymbol{\xi}_i, \boldsymbol{\xi}_j)$, is defined as the dot product of the gradients of the loss function $\mathcal{J}(\cdot)$ with respect to the model parameters, evaluated at $\boldsymbol{\theta}^t$:*

$$Inf_t(\boldsymbol{\xi}_i, \boldsymbol{\xi}_j) = \nabla\mathcal{J}(\boldsymbol{\theta}^t; \boldsymbol{\xi}_i) \cdot \nabla\mathcal{J}(\boldsymbol{\theta}^t; \boldsymbol{\xi}_j). \quad (4)$$

*This quantity measures the first-order effect of updating the model with data $\boldsymbol{\xi}_i$ on the loss of data $\boldsymbol{\xi}_j$, capturing the similarity in their training dynamics.*

The gradients for each data point reflect the average gradients of all tokens within that data. Previous studies have observed that the gradient norm is inversely correlated with response length (Liu et al., 2025b; Xia et al., 2024a). Using only the inner product of gradients between two data points may bias the data selector toward shorter sequences. To address this issue, some works (Wang et al., 2020; Xia et al., 2024a) employ the cosine similarity instead, but they still suffer from performance degradation when selecting data for post-training LLMs.

## 4.3 IMPROVING GRADIENT ALIGNMENT WITH LEARNING POTENTIAL

**Motivation:** Based on the preceding analysis, the post-training dynamics of large language models reveal two critical limitations when using the cosine similarity of data gradients as a selection criterion: (1) **Loss of Magnitude Information.** By normalizing the gradients, the cosine similarity focuses exclusively on their directional alignment, thereby discarding magnitude information. In post-training LLMs, the gradient magnitude often indicates a data point's influence on model updates, which is essential for effective policy optimization. Ignoring this aspect prevents the cosine similarity from prioritizing data that could drive more substantial improvements in model performance. (2) **Failure to Capture Learning Potential.** The cosine similarity does not account for the learning potential of data. Even if two data points exhibit high directional similarity, their utility may be limited if they

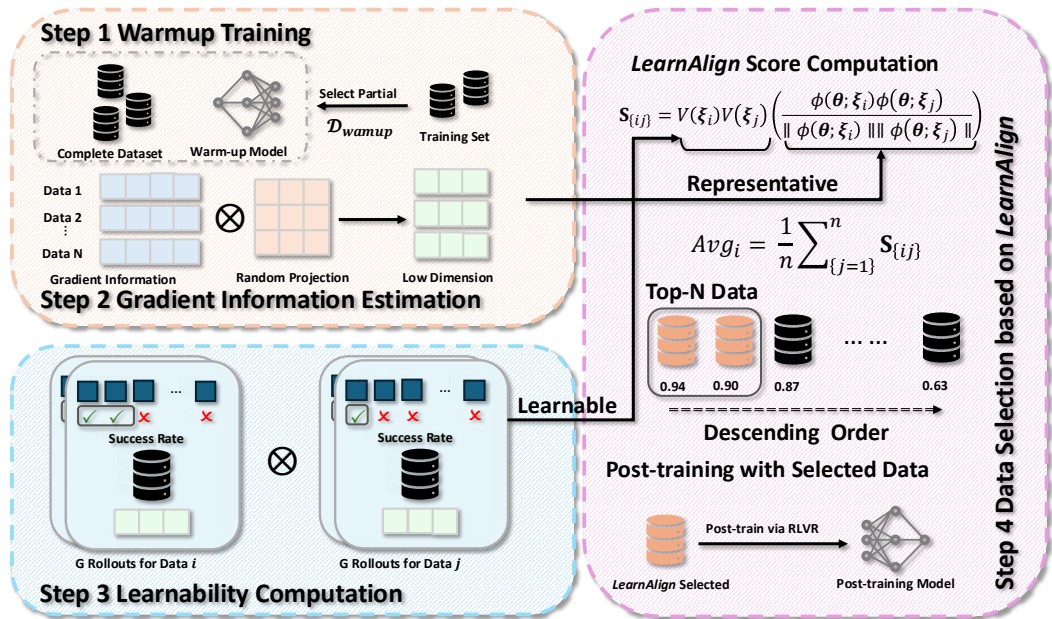

Figure 2: The procedure of the proposed selection method based on improved gradient alignment. We obtain the gradient information and learnability of each data point through Steps 1-3 and then select data for subsequent training according to datapoint-wise *LearnAlign* score in Step 4.

are either too easy (success rate $p \approx 1$) or too difficult (success rate $p \approx 0$) for the current policy, leading to suboptimal data selection. This limitation aligns with the theory of the Zone of Proximal Development (ZPD) (Chaiklin et al., 2003), which suggests that effective learning occurs when tasks are of moderate difficulty—neither too challenging nor too simple—for the learner (e.g., an LLM).

To address the aforementioned limitations, we introduce a data learnability metric based on the success rate $p$, drawing inspiration from prior work to account for both the learning potential and the magnitude of the data (Florensa et al., 2018; Tzannetos et al., 2023; Foster & Foerster, 2025).

**Definition 4.2** (Data Learnability). *Consider a sample $\boldsymbol{\xi}$ evaluated by an LLM policy $\pi_{\boldsymbol{\theta}}$. Let $p \in [0, 1]$ represent the success rate, defined as the fraction of successful outcomes for the query $\boldsymbol{\xi}$ across $G$ rollouts, where $p$ reflects the probability of a successful learning outcome. The data learnability of data $\boldsymbol{\xi}$, denoted $V(\boldsymbol{\xi})$, is defined as:*

$$V(\boldsymbol{\xi}) = p(1 - p),$$

*where $1 - p$ represents the potential for improvement, and $p(1 - p)$ quantifies the expected learnability of data. This measure captures the sample's utility for enhancing the policy $\pi_{\boldsymbol{\theta}}$, reaching its maximum when $p = 0.5$, indicating a sample at the boundary of the policy's current capability. Besides, the detailed justification for the data learnability can be found in Appendix C.*

Built upon the above motivation and our definition of data learnability, we first define a new learnability-weighted gradient vector for each data point $\boldsymbol{\xi}_i$ as:

$$\mathbf{V}(\boldsymbol{\xi}_i) = \frac{\nabla \mathcal{J}(\boldsymbol{\theta}; \boldsymbol{\xi}_i)}{\|\nabla \mathcal{J}(\boldsymbol{\theta}; \boldsymbol{\xi}_i)\|} \cdot V(\boldsymbol{\xi}_i), \tag{5}$$

where the first term is the unit gradient vector and $V(\boldsymbol{\xi}_i)$ is the learnability score (Definition 4.2). Using these vectors, we can then compute the *LearnAlign* Score between two data points $\boldsymbol{\xi}_i$ and $\boldsymbol{\xi}_j$ as

$$LearnAlign(\boldsymbol{\xi}_i, \boldsymbol{\xi}_j) = \mathbf{V}(\boldsymbol{\xi}_i) \cdot \mathbf{V}(\boldsymbol{\xi}_j) = V(\boldsymbol{\xi}_i)V(\boldsymbol{\xi}_j)\frac{\nabla \mathcal{J}(\boldsymbol{\theta}; \boldsymbol{\xi}_i)^{\top} \nabla \mathcal{J}(\boldsymbol{\theta}; \boldsymbol{\xi}_j)}{\|\nabla \mathcal{J}(\boldsymbol{\theta}; \boldsymbol{\xi}_i)\|\|\nabla \mathcal{J}(\boldsymbol{\theta}; \boldsymbol{\xi}_j)\|}. \tag{6}$$

This formulation leverages the learnability of each data point to weight the gradient inner product by the learning potential, thus reducing the tendency to favor shorter sequences.

### 4.4 DATA SELECTION FOR RLVR POST-TRAINING

As shown in Figure 2, the procedure to select suitable data for LLM RL consists of four steps, where we elaborate *LearnAlign* from step 1 to step 4 in detail.

**Step 1. Warmup Training:** Initially, we randomly select a small subset $\mathcal{D}_{\text{warmup}} \subset \mathcal{D}_{\text{train}}$ from the training dataset to perform warmup training on the policy model $\pi_\theta$. This step ensures a more stable and accurate gradient estimation, resulting in a warm-up model $\boldsymbol{\theta}_s$.

**Step 2. Gradient Information Estimation:** Additionally, we can derive the original gradient information from the model $\boldsymbol{\theta}$ checkpoint during the warmup phase of RL-based LLM post-training (e.g., GRPO) as follows:

$$\nabla_{\boldsymbol{\theta}} \mathcal{J}_{\text{GRPO}}(\boldsymbol{\theta}) = \mathbb{E}_{(q,a)\sim\mathcal{P}_q,\{o_i\}_{i=1}^G \sim \pi_{\boldsymbol{\theta}_{old}}(o|q)} \left\{ \frac{1}{G} \sum_{i=1}^{G} \frac{1}{|o_i|} \sum_{t=1}^{|o_i|} G(q,a,t,\pi_{\boldsymbol{\theta}}) \nabla_{\boldsymbol{\theta}} \log \pi_{\boldsymbol{\theta}}(o_{i,t}|q,o_{i,<t}) \right\},$$

(7)

where $G(q,a,t,\pi_{\boldsymbol{\theta}})$ denotes the gradient coefficient $\hat{A}_{i,t} + \beta \left( \frac{\pi_{\text{ref}}(o_{i,t}|q,o_{i,<t})}{\pi_{\boldsymbol{\theta}}(o_{i,t}|q,o_{i,<t})} - 1 \right)$, $\hat{A}_{i,t}$ is computed as GRPO. Since this gradient has nearly the same dimensions as the original model, it is computationally complex. Following prior work, we apply a random projection $\Gamma$ to the gradient information for each data point (Johnson et al., 1984; Xia et al., 2024a). So we can get a low-dimensional gradient-related information denoted as $\phi(\boldsymbol{\theta}; \boldsymbol{\xi}) = \Gamma^\top \nabla \mathcal{J}_{\text{GRPO}}(\boldsymbol{\theta}; \boldsymbol{\xi})$.

**Step 3. Learnability Computation:** We first sample $G$ rollouts for each question and compute the success rate of question $i$ based on the ground truth answer $\mathbf{y}^*$ and the generated answers $\mathbf{y}$ across these $G$ rollouts. The success rate $p$ is calculated as $p = \frac{1}{G} \sum_{g=1}^{G} \mathbb{I}(\mathbf{y}_g = \mathbf{y}^*)$, where $\mathbb{I}$ is the indicator function. Following Definition 4.2, we can get the learnability $V(\xi_i)$ for each data $i$.

**Step 4. Data Selection based on *LearnAlign*:** Based on the projected gradient from the warmed-up model $\boldsymbol{\theta}_s$, we can rewrite the *LearnAlign* Score between two data $\xi_i$ and $\xi_j$ as:

$$S_{ij} = V(\boldsymbol{\xi}_i) V(\boldsymbol{\xi}_j) \left( \frac{\phi(\boldsymbol{\theta}; \boldsymbol{\xi}_i) \phi(\boldsymbol{\theta}; \boldsymbol{\xi}_j)}{\|\phi(\boldsymbol{\theta}; \boldsymbol{\xi}_i)\| \|\phi(\boldsymbol{\theta}; \boldsymbol{\xi}_j)\|} \right). \tag{8}$$

So we can get a $n \times n$ *LearnAlign* Score Matrix $\mathbf{S}$ (where $|\mathcal{D}_{\text{train}}| = n$), capturing the pairwise relation among all data points in the training dataset. Using the *LearnAlign* Score Matrix $\mathbf{S}$, we select the top-N data. For each data $\xi_i$, the average *LearnAlign* Score across its row as $\text{Avg}_i = \frac{1}{n} \sum_{j=1}^{n} S_{ij}$, where $S_{ij}$ represents the pairwise alignment scores for all $j$ (including $j = i$) and $|D_{\text{train}}| = n$. These average scores are then sorted in descending order, and the top-N samples with the highest averages are selected, ensuring the chosen data exhibit the strongest learnability within the training dataset.

## 5 EXPERIMENTS

We first introduce the experimental setup (Section 5.1) of *LearnAlign*, and then we present the main results (Section 5.2) on the five benchmarks with some key observations. Moreover, we give some discussions (Section 5.3), and complexity analysis (Section 5.4) about our methods.

### 5.1 EXPERIMENTAL SETUP

**Settings:** We validate the effectiveness of our algorithm under two primary configurations: **(1)** We train models on subsets of the GSM8K (Cobbe et al., 2021) training set with varying sizes: 100, 500, 1,000, and 2,000 samples. The base policy model is Qwen2.5-1.5B-Instruct, and evaluation is performed on the GSM8K test set, with greedy decoding used during the inference stage, and the pass@1 accuracy is reported. **(2)** We train on 1,000 samples from the DAPO-MATH-17K dataset (Yu et al., 2025) training set using Qwen2.5-3B and Qwen2.5-7B as the initial policy model. Evaluation is conducted on both math reasoning benchmarks (GSM8K (Cobbe et al., 2021), MATH500 (Hendrycks et al., 2021a), AMC2023 (AMC, 2023), and AIME2024 (AIM, 2024)) and one code generation

Table 1: Comparison of data selection methods on GSM8K test set. We train Qwen2.5-1.5B-Instruct on the GSM8K training selected subset.

| Data Selection Method | Selected Data Size | | | |
|---|---|---|---|---|
| | **100** | **500** | **1,000** | **2,000** |
| Qwen2.5-1.5B-Instruct | 55.7 | | | |
| Qwen2.5-1.5B-Instruct-FULL | 77.0 | | | |
| Random Sampling | 73.1 | 75.1 | 75.6 | 75.5 |
| PPL-Top (Laurençon et al., 2022) | 72.5 | 75.8 | 74.6 | 75.2 |
| PPL-Middle (Ankner et al., 2024) | 72.8 | 74.7 | 75.0 | 74.2 |
| IFD (Li et al., 2023a) | 72.0 | 76.0 | 75.6 | 75.4 |
| Token Length (Xia et al., 2024b) | 72.3 | 74.4 | 76.2 | 75.6 |
| SelectIT (Liu et al., 2024a) | 72.8 | 75.7 | 75.6 | 75.5 |
| LIMR (Li et al., 2025) | 74.2 | 76.2 | 76.1 | 76.7 |
| *LearnAlign* | **74.8** | **76.4** | **77.5** | **78.3** |

benchmark (CRUX (Gu et al., 2024)). For GSM8K, MATH500, and CRUX, we report the pass@1 accuracy; for AMC2023, we report avg@8 as the metric; for AIME2024, we report the pass@8 accuracy. The evaluation temperature is set to 0.8, and the tokp is set to 0.95.

**Implementation Details:**    In these experimental settings, for the training hyperparameters, during exploration, we generated 8 rollouts per sample at a temperature of $1.0$; the learning rate was set to $1.0 \times 10^{-6}$; the KL coefficient $\beta$ was fixed at $0.04$; and the clipping parameter $\epsilon$ was set to $0.2$. The batch size is set to 48 for GSM8K and 64 for DAPO-MATH-17K. We follow (Xia et al., 2024a) for the projection of gradients and use 300 and 1000 samples for warmup training in GSM8K and DAPO-MATH-17K, respectively. For DAPO-MATH-17K, inspired by (Lin et al., 2025), we calculate the gradient of one correct rollout for each sample. Additional details are provided in Appendix B.1.

**Baselines:**    We compared *LearnAlign* with several baselines: **Random Sampling**, **PPL-Top** (Laurençon et al., 2022),  **PPL-Middle** (Ankner et al., 2024) **IFD** (Li et al., 2023a), **Token Length** (Xia et al., 2024b),**SelectIT** (Liu et al., 2024a), and **LIMR** (Li et al., 2025). For GSM8K, we utilize the official solutions in training data as responses to calculate the above metrics. For DAPO-MATH-17K, we make the warmed-up model to generate one response for each problem to conduct their selection. More details about the baselines can refer to Appendix B.2.

## 5.2    MAIN RESULTS

Table 1 presents the evaluation results of training models on the GSM8K dataset with varying selected data sizes. Table 2 shows the evaluation results of training models on the DAPO-MATH-17K dataset. From these results, we have the following key observations:

**Key Observation 1: Traditional SFT data selection methods fall short in the RLVR paradigm for the post-training phase of LLMs.**    on the one hand, as shown in Table 1, when the official solutions of the training data are applied as the responses in data selection, traditional SFT approaches show limited and inconsistent effectiveness when applied to RL post-training. For example, Token Length performs well at 1,000 samples (76.2%) but drops at 2,000 samples (75.6%). On the other hand, as shown in Table 2, when the rollouts of the warmed-up model are generated for data selection, PPL-Top are slightly higher than Random Sampling on average. Note that none of these baselines consistently outperforms random sampling across the five benchmarks. Such suboptimal performance of SFT data selection methods may stem from a misalignment between SFT and RL objectives. SFT post-training aims to maximize the likelihood of target outputs, where harder examples identified by those methods are often more valuable (assuming they are not noisy). RL post-training optimizes for reward maximization, requiring the difficulty to match the model's current capability.

**Key Observation 2: *LearnAlign* achieves minor performance degradation or superior performance while requiring only a fraction of the training data.**    As shown in Table 1, our approach

Table 2: Comparison of data selection methods on four math benchmarks (GSM8K, MATH500, AMC2023, AIME2024) and one code benchmark (CRUX). We train Qwen2.5-3B and Qwen2.5-7B on the DAPO-MATH-17K selected subset with 1,000 data points.

| Data Selection Method | GSM8K | MATH500 | AMC2023 | AIME2024 | CRUX | Avg. |
|---|---|---|---|---|---|---|
| Qwen2.5-3B | 20.1 | 52.2 | 8.1 | 3.3 | 14.6 | 19.7 |
| Qwen2.5-3B-FULL | 83.6 | 65.8 | 31.0 | 20.0 | 24.3 | 44.9 |
| Random Sampling | 70.5 | 53.4 | 19.4 | 13.3 | 17.4 | 34.8 |
| PPL-Top (Laurençon et al., 2022) | 71.3 | 57.8 | 23.0 | 13.3 | 16.1 | 36.3 |
| PPL-Middle (Ankner et al., 2024) | 70.1 | 54.0 | 24.1 | 10.0 | 18.1 | 35.3 |
| IFD (Li et al., 2023a) | 70.9 | 54.4 | 23.3 | 6.7 | 15.4 | 34.1 |
| Token Length (Xia et al., 2024b) | 35.7 | 50.4 | 18.5 | 16.7 | 15.3 | 27.3 |
| SelectIT (Liu et al., 2024a) | 70.1 | 60.2 | 25.2 | 16.7 | 17.8 | 38.0 |
| LIMR (Li et al., 2025) | 74.0 | 55.6 | 25.6 | **23.3** | 16.5 | 39.0 |
| *LearnAlign* | **79.3** | **60.2** | **28.3** | **23.3** | **21.0** | **42.4** |
| Qwen2.5-7B | 26.4 | 67.2 | 18.1 | 16.7 | 25.1 | 30.7 |
| Qwen2.5-7B-FULL | 89.8 | 76.4 | 47.0 | 30.0 | 51.1 | 58.9 |
| Random Sampling | 81.1 | 65.0 | 30.1 | 23.3 | 40.8 | 48.1 |
| PPL-Top (Laurençon et al., 2022) | 87.7 | 65.4 | 28.0 | 20.0 | 42.5 | 48.7 |
| PPL-Middle (Ankner et al., 2024) | 85.1 | 64.4 | 27.3 | 16.7 | 43.3 | 47.4 |
| IFD (Li et al., 2023a) | 79.4 | 58.6 | 29.8 | 13.3 | 34.9 | 43.2 |
| Token Length (Xia et al., 2024b) | 81.4 | 62.2 | 31.0 | 20.0 | 38.1 | 46.5 |
| SelectIT (Liu et al., 2024a) | 85.4 | 67.0 | 32.7 | 26.7 | 41.5 | 50.7 |
| LIMR (Li et al., 2025) | 84.2 | 61.6 | 27.1 | 16.7 | 39.9 | 45.9 |
| *LearnAlign* | **88.3** | **70.4** | **35.4** | **30.0** | **44.0** | **54.6** |

consistently outperforms baselines at every data scale, achieving comparable or superior performance to full-data training with a small amount of the data. Specifically, With 1,000 samples ($\approx 13.4\%$ of full data), *LearnAlign* reaches 77.5%, already matching the full-data baseline (77.0%). With 2,000 samples ($\approx 26.8\%$ of full data), the proposed method significantly surpasses full-data training (78.3% vs. 77.0%). Besides, with fewer samples (e.g., 100 and 500), the proposed data selection method can largely improve the base model (55.7%) and even exceed other baselines with more samples, proving that smart selection is better than brute-force scaling, i.e., RL post-training with a carefully curated seed set can rapidly unlock a pretrained model's reasoning ability (Li et al., 2025).

**Key Observation 3: *LearnAlign* shows consistent effectiveness across various settings.** As shown in Table 1 and Table 2, our proposed data selection method demonstrates consistent SOTA performance not only on in-distribution (GSM8K, MATH500) but also on out-of-distribution (AMC2023, AIME2024) test sets, and it even generalizes well on the code domain benchmark (CRUX). In addition, as shown in Appendix K, *LearnAlign* boosts the performance of RL post-training in the staged setting. These results show that it can be effectively applied in various settings by considering learnability and alignment.

## 5.3 DISCUSSIONS

**Response-length bias issue:** Similar to SFT, sequence-level policy gradients require averaging across tokens within a sequence. As shown in Figure 3a, the gradient norm exhibits an inverse correlation with response length, introducing a systematic bias. Consequently, as shown in Figure 3b, compared with *LearnAlign*, which replaces gradient norms with success-rate–based learnability, the data selected by vanilla gradient matching yields much shorter responses and lower performance. Given that incorrect responses may lead to longer outputs, *LearnAlign* selects data with more moderate response lengths between vanilla gradient and random, and achieves higher average performance. Therefore, success-rate-based learnability serves as a more suitable indicator than raw gradient norms.

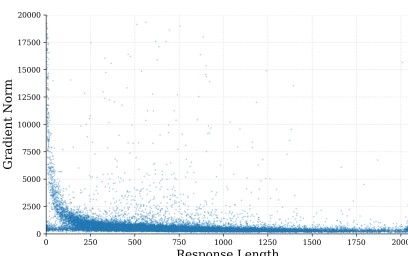

(a) Gradient norms of examples negatively correlate with the length of the response.

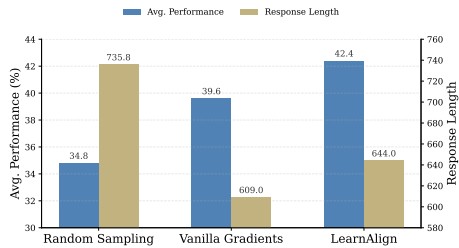

(b) Vanilla gradients method selects shorter examples and leads to worse performance.

Figure 3: Analysis of response length and gradient-based example selection.

Table 3: Ablation study of our method with Qwen2.5-1.5B-Instruct and Qwen2.5-3B model.

| Model | Qwen2.5-1.5B-Instruct | | Qwen2.5-3B | | |
|---|---|---|---|---|---|
| Data Size
Benchmark | 1,000
GSM8K | 2,000
GSM8K | 1,000
GSM8K | 1,000
MATH500 | 1,000
AMC2023 |
| *LearnAlign* | **77.5** | **78.3** | **79.3** | **60.2** | **28.3** |
| w/o warmup training | 76.6 | 76.6 | 76.7 | 58.2 | 26.1 |
| w/o the data learnability | 75.6 | 76.7 | 77.5 | 58.4 | 28.3 |
| w/ feature similarity | 75.7 | 76.6 | 79.1 | 57.6 | 27.5 |

**Ablation studies:** We conducted three ablation studies on the GSM8K dataset with 1,000 and 2,000 problems, and the DAPO-MATH-17K dataset with 1,000 problems: (1) removing the warmup phase; (2) omitting the learnability metric; and (3) replacing the cosine similarity between gradients with a feature-similarity measure (Ivison et al., 2025). As shown in Table 3, the removal of any single component leads to a decline in performance. It indicates that the warmup phase, the learnability metric, and gradient similarities each make a significant contribution to letting the data selection method capture the model's current capability. These findings align with the extended results in Appendix F, further confirming that both warmup training and data learnability play essential roles in the effectiveness of the proposed method.

**More training steps discussion:** To examine whether the selected subset constrains the final achievable performance, we train the LearnAlign-selected data with more steps from 250 to 2000. As shown in Table 5, training with more steps on the selected subset reaches the FULL-dataset performance and even surpasses it.

**Convergence behavior analysis:** As shown in Figure 4, FULL training peaks at 63.12% validation accuracy at step 640, whereas LearnAlign reaches the same accuracy by step 440, using 31% fewer steps. This indicates substantially faster convergence under identical budgets. LearnAlign then surpasses FULL's peak, achieving 64.22% at step 1040, after which its curve remains stable with a smoother plateau than FULL.

## 5.4 COMPLEXITY ANALYSIS

Let $n = |\mathcal{D}_{\text{train}}|$, $m = |\mathcal{D}_{\text{warmup}}| \ll n$, and $d$ be the projected gradient dimension. Let $C_{\nabla \mathcal{J}}$ and $C_{\text{gen}}$ denote the time cost of computing one gradient and generating one rollout, respectively. The data selection includes four steps: **(1)** RL fine-tuning on $\mathcal{D}_{\text{warmup}}$ to obtain $\boldsymbol{\theta}_s$: time $\mathcal{O}(mC_{\nabla \mathcal{J}})$, space $\mathcal{O}(\dim(\boldsymbol{\theta}))$. **(2)** Computing GRPO gradients for each $\boldsymbol{\xi} \in \mathcal{D}_{\text{train}}$ and projecting to $\phi(\boldsymbol{\theta}_s; \boldsymbol{\xi}) \in \mathbb{R}^d$: time $\mathcal{O}(nC_{\nabla \mathcal{J}})$, space $\mathcal{O}(nd)$. **(3)** Generating $G$ rollouts per sample and computing Learnability: time $\mathcal{O}(nGC_{\text{gen}})$, space $\mathcal{O}(n)$. **(4)** Constructing the pairwise score matrix $\mathbf{S} \in \mathbb{R}^{n \times n}$ and averaging rows to select top-$N$: time $\mathcal{O}(n^2 d)$, space $\mathcal{O}(n^2)$. Note that the alternative data selection method LIMR (Li et al., 2025) requires multi-epoch training on the full dataset. As shown in Table 4, our approach offers a more practical solution for RLVR post-training. For the time-cost analysis of all steps and a detailed discussion, please refer to Appendix G.

Table 4: Comparison of time cost for training Qwen2.5-3B on the DAPO-MATH-17K selected 1,000 subset with different methods. Time is reported in hours on a single H100 GPU. In the DAPO-MATH-17K experiments, inspired by (Lin et al., 2025), we calculate the gradient of one correct rollout for each sample. * means that we calculate the gradients of all rollouts for each sample.

| Method | Data Selection Time | Training Time | Speedup | Avg. performance |
|---|---|---|---|---|
| FULL | - | 42.3h | x1.00 | 44.9 |
| LIMR | 42.3h | 2.4h | x0.95 | 39.0 |
| *LearnAlign* | 8.9h | 2.4h | x3.74 | 42.4 |
| *LearnAlign** | 22.8h | 2.4h | x1.68 | 43.3 |

Table 5: Performance of our method with more training steps. * The FULL method on Qwen2.5-3B uses 2,174 training steps, and when training on Qwen2.5-7B, it uses 1,000 training steps with a training batch size of 256 to support long-time training and prevent training crashes.

| Method | Qwen2.5-3B | | | Qwen2.5-7B | | |
|---|---|---|---|---|---|---|
| | GSM8K | MATH500 | AMC2023 | GSM8K | MATH500 | AMC2023 |
| FULL* | 83.6 | 65.8 | 31.0 | 90.0 | **77.6** | 47.3 |
| *LearnAlign* (250 steps) | 79.3 | 60.2 | 28.3 | 88.3 | 70.4 | 35.4 |
| *LearnAlign* (500 steps) | 80.7 | 63.4 | 31.5 | 89.0 | 75.3 | 43.8 |
| *LearnAlign* (1,000 steps) | 82.9 | 64.6 | 35.2 | **90.4** | 76.7 | **48.6** |
| *LearnAlign* (2,000 steps) | **83.8** | **67.8** | **36.9** | - | - | - |

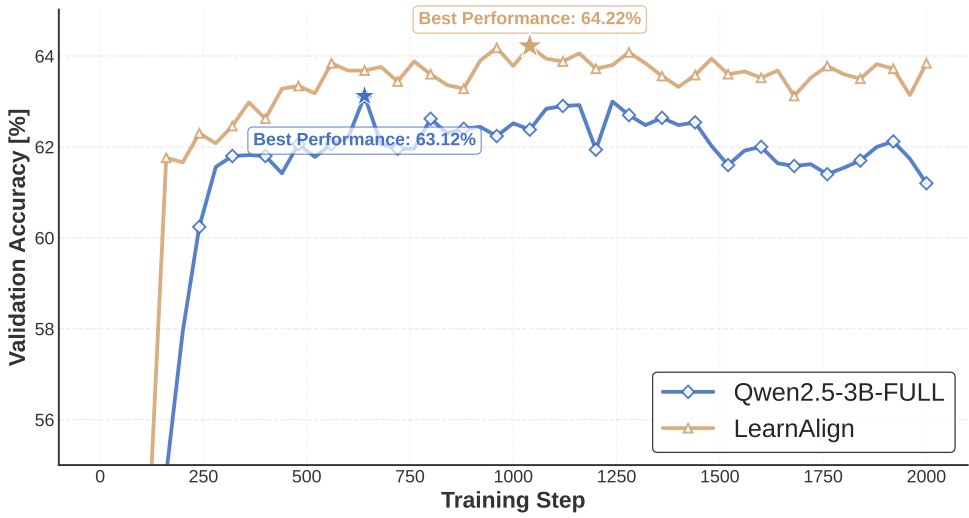

Figure 4: Validation accuracy vs. training step for *LearnAlign* and the FULL method with Qwen2.5-3B. The validation is conducted on the validation set of the MATH dataset (Hendrycks et al., 2021b).

## 6   CONCLUSION

In this study, we propose a novel data selection framework for reinforcement learning post-training of large language models, driven by a gradient-alignment method. Building upon policy-gradient direction alignment, our framework introduces a success-rate-based learnability score to mitigate response-length bias and efficiently identify a compact subset of reasoning examples. Experiments on the five benchmarks demonstrate that, with only approximately 1,000 samples (less than 15% of the full training set), our method matches or surpasses the performance of full-data training on both in-distribution and out-of-distribution tasks.

ETHICS STATEMENT

This paper raises no ethical concerns. The study does not involve human subjects, dataset release practices, potentially harmful insights, methodologies, or applications. Additionally, it is free from conflicts of interest, sponsorship issues, discrimination, bias, fairness concerns, privacy or security risks, legal compliance issues, or research integrity concerns.

REPRODUCIBILITY STATEMENT

To ensure the reproducibility of all experiments, we have provided detailed hyperparameters for all experimental results. Due to privacy considerations, we will share an anonymized link to the source code and instructions during the discussion phase, accessible exclusively to reviewers.

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

# APPENDIX

## A    LIMITATIONS

Due to limited GPU resources, we only evaluate the effectiveness of data selection methods on relatively small-scale models (1.5B, 3B and 7B models) and datasets. Specifically, our current assessment of the proposed method's effectiveness focuses on math reasoning datasets, including GSM8K and DAPO-MATH-17k. In the future, we plan to evaluate it on larger models and diverse datasets. We believe this work establishes an effective paradigm for data-efficient RL fine-tuning. Future research directions may encompass the extension to a broader range of task domains, the integration of dynamic curricula with adaptive selection strategies, and the pursuit of alignment with out-of-distribution data.

## B    ADDITIONAL EXPERIMENTAL DETAILS

### B.1    HYPERPARAMETES AND PROMPT

For additional experimental hyperparameters, please refer to Table 6. The prompts used for GSM8K and DAPO-MATH-17K are as follows:

> **The System Prompt for GSM8K:**
>
> A conversation between User and Assistant. The user asks a question, and the Assistant solves it. The assistant first thinks about the reasoning process in the mind and then provides the user with the answer. The reasoning process and answer are enclosed within <think> </think> and <answer> </answer> tags, respectively, i.e., <think> reasoning process here </think> <answer> answer here </answer>.

> **The System Prompt for DAPO-MATH-17K:**
>
> Let's think step by step and output the final answer within \boxed{}.

### B.2    DETAILED COMPARED METHODS

In this section, we detail the baseline methods compared with *LearnAlign*. **Random Sampling:** We randomly select a portion of all the datasets as the training set data. **PPL-Top** (Laurençon et al., 2022) and **PPL-Middle** (Ankner et al., 2024) all based on the perplexity calculated by Eq.(9):

$$\text{PPL}(\boldsymbol{\xi}) = \exp\left(-\frac{1}{T}\sum_{t=1}^{T}\log \pi_{\boldsymbol{\theta}}(y_t|\boldsymbol{\xi}_{0:t-1})\right), \tag{9}$$

where PPL-Top selects data with the top perplexity, while PPL-Middle selects the data with the middle perplexity. Furthermore, **Instruction-Following Difficulty (IFD)** (Li et al., 2023a) quantifies the inherent difficulty of an instruction-answer pair for a Large Language Model (LLM). It is calculated as the ratio between the direct answer score $s_{\boldsymbol{\theta}}(o)$ and the conditioned answer score $s_{\boldsymbol{\theta}}(o|q)$. Direct answer score $s_{\boldsymbol{\theta}}(o)$ is the averaged cross-entropy loss of generating the answer $o$ without any instructional context. At the same time, conditioned answer score $s_{\boldsymbol{\theta}}(o|q)$ is the averaged

Table 6: More detailed experimental parameter setting.

| Training Dataset | GSM8K | DAPO-MATH-17K |
|---|---|---|
| **Training Configuration** | | |
| Train Batch Size | 48 | 64 |
| Max Prompt Length | 512 | 512 |
| Max Response Length | 1024 | 2048 |
| Train epochs | 2 | 2 |
| Clip Ratio | 0.2 | 0.2 |
| **Optimizer Parameters** | | |
| Optimizer | AdamW ($\beta_1 = 0.9, \beta_2 = 0.999, \epsilon = 10^{-8}$) | AdamW ($\beta_1 = 0.9, \beta_2 = 0.999, \epsilon = 10^{-8}$) |
| Learning Rate | 1e-06 | 1e-06 |
| Warmup Style | Cosine | Cosine |
| Warmup Steps Ratio | 0.1 | 0.1 |
| KL Loss Coefficient | 0.04 | 0.04 |
| **Temperature** | | |
| Training Temperature | 1.0 | 1.0 |
| Evaluation Temperature | 0 | 0.8 |

cross-entropy loss of generating the ground-truth answer $o$ given the instruction $q$. The IFD is then calculated as:

$$\text{IFD}_{\boldsymbol{\theta}} = \frac{s_{\boldsymbol{\theta}}(o|q)}{s_{\boldsymbol{\theta}}(o)}, \tag{10}$$

where a higher IFD score indicates that the instruction provides less benefit to the response generation.

**Token Length** (Xia et al., 2024b) quantifies the value of a sample based on its token count. We calculate the total token length by combining the tokens from both the question and the answer. **SelectIT** (Liu et al., 2024a) harnesses the inherent uncertainty within the LLMs. This approach utilizes a multi-granularity self-reflection mechanism, seamlessly integrating token-level, sentence-level, and parameter-weighted model-level uncertainty analyses to evaluate and rank the quality of instruction data. **LIMR** (Li et al., 2025) measures the learning impact of each training sample by its alignment with the overall learning trajectory of the model.

## C  THEORETICAL MOTIVATION FOR THE LEARNABILITY METRIC

Although the proposed learnability metric $p(1 - p)$ may appear simple, it is in fact a theoretically grounded formulation for modeling learnability under Bernoulli feedback in RLVR.

First, the success rate $p$ measures how often the model receives informative positive trajectories revealing correct behavior, while $1 - p$ captures the remaining room for improvement. A sample provides a useful learning signal only when both conditions co-exist, and thus their product $p(1 - p)$ can represent the expected improvement that a data point will provide (Florensa et al., 2018; Tzannetos et al., 2023).

Second, $p(1 - p)$ is precisely the variance of Bernoulli accuracy rewards. Recent theoretical analyses (Razin et al., 2025; Bae et al., 2025) show that the reward variance lower-bounds the KL divergence between the initial and the optimal model, making it an effective statistical quantity reflecting the gradient informativeness of a sample.

Third, this quadratic form is not arbitrary: it is the unique smooth, symmetric, unimodal function that (i) peaks at intermediate difficulty, (ii) vanishes at $p = 1$, and (iii) aligns with Fisher information based measures of sample utility (MacKay, 1992). Alternative function choices fail to satisfy these properties or lack comparable theoretical interpretability.

Last but not least, as shown in Appendix D, for a fixed query $\xi$ and model $\theta$, the gradient magnitude is positively proportional to $p(1 - p)$. It indicates that $p(1 - p)$ can represent the information about the gradient magnitude without the issue of response-length bias.

Overall, $p(1 - p)$ is a principled, theoretically grounded, and empirically supported metric for modeling learnability.

# D THEORETICAL ANALYSIS OF THE LEARNABILITY–GRADIENT RELATIONSHIP

Here, we prove a theorem to show the gradient magnitude is positively proportional to $p(1-p)$. Given a prompt $\xi$ and a response $y \in \mathcal{Y}$. The policy $\pi_\theta(y \mid \xi)$ has logits $z_{\xi,y}(\theta)$, which are functions of the parameters $\theta$; for simplicity, we denote them as $z_{\xi,y}$ in the following. Under this notation, the policy satisfies

$$\pi_\theta(y \mid \xi) = \frac{\exp(z_{\xi,y})}{\sum_{y' \in \mathcal{Y}} \exp(z_{\xi,y'})}.$$

Let $y^*$ be the unique correct action with success probability $p := \pi_\theta(y^* \mid \xi)$, binary reward $r(y) = \mathbf{1}[y = y^*]$, and baseline $b(\xi) := \mathbb{E}_{y \sim \pi_\theta(\cdot \mid \xi)}[r(y)] = p$. For simplicity, assume a single incorrect action $\bar{y} \neq y^*$, with probability $1 - p = \pi_\theta(\bar{y} \mid \xi)$.

**Theorem D.1** (Gradient Magnitude Factorization). *For the one-correct-answer setting with binary reward, the policy gradient for a sample $\xi$ can be written as*

$$\nabla_\theta \mathcal{J}(\theta; \xi) = p(1-p)\, \mathbf{d}(\xi, \theta),$$

*for the direction vector $\mathbf{d}(\xi, \theta) \in \mathbb{R}^{\dim(\theta)}$. Consequently, for fixed $\xi$ and $\theta$,*

$$\|\nabla_\theta \mathcal{J}(\theta; \xi)\| \propto p(1-p),$$

*i.e., the gradient magnitude is positively proportional to $p(1-p)$.*

*Proof.* The advantage for an action can be expressed as:

$$A(\xi, y) = r(y) - b(\xi), \qquad b(\xi) = p,$$

where the baseline is chosen to be the constant $b(\xi) = p$. Consequently,

$$A(\xi, y^*) = 1 - p, \qquad A(\xi, \bar{y}) = -p.$$

Consider the expected advantage under the policy $\pi_\theta$:

$$\mathcal{J}(\theta; \xi) := \mathbb{E}_{y \sim \pi_\theta(\cdot \mid \xi)}[A(\xi, y)]$$

The policy gradient is then given by the standard identity:

$$\nabla_\theta \mathcal{J}(\theta; \xi) = \mathbb{E}_{y \sim \pi_\theta(\cdot \mid \xi)}\big[A(\xi, y)\, \nabla_\theta \log \pi_\theta(y \mid \xi)\big].$$

For a softmax policy parameterized by logits $z_{\xi,y}$, the score function satisfies:

$$\frac{\partial \log \pi_\theta(y' \mid \xi)}{\partial z_{\xi,y}} = \mathbf{1}[y' = y] - \pi_\theta(y \mid \xi).$$

Differentiating $\mathcal{J}(\theta; \xi)$ with respect to a specific logit $z_{\xi,y}$ therefore yields:

$$\frac{\partial \mathcal{J}(\theta; \xi)}{\partial z_{\xi,y}} = \pi_\theta(y \mid \xi)A(\xi, y) - \pi_\theta(y \mid \xi)\, \mathbb{E}_{y' \sim \pi_\theta}[A(\xi, y')].$$

Since the baseline is the expected reward probability,

$$\mathbb{E}_{y' \sim \pi_\theta}[A(\xi, y')] = \mathbb{E}[r(y') - p] = p - p = 0,$$

we obtain the simplified logit gradient:

$$\frac{\partial \mathcal{J}(\theta; \xi)}{\partial z_{\xi,y}} = \pi_\theta(y \mid \xi)A(\xi, y).$$

The corresponding logit update is:

$$\Delta z_{\xi,y} \propto \pi_\theta(y \mid \xi)A(\xi, y),$$

Substituting the two possible actions. Let $\pi_\theta(y^* \mid \xi) = p$, then:

$$\Delta z_{\xi,y^*} \propto p(1-p), \qquad \Delta z_{\xi,\bar{y}} \propto -(1-p)p.$$

By the chain rule, the full parameter gradient satisfies:

$$\nabla_\theta \mathcal{J}(\theta; \xi) \propto \frac{\partial z_{\xi,y^*}}{\partial \theta} \, p(1-p) + \frac{\partial z_{\xi,\bar{y}}}{\partial \theta} \left[-p(1-p)\right]$$

$$= p(1-p) \left( \frac{\partial z_{\xi,y^*}}{\partial \theta} - \frac{\partial z_{\xi,\bar{y}}}{\partial \theta} \right).$$

Define

$$\mathbf{d}(\xi,\theta) := \frac{\partial z_{\xi,y^*}}{\partial \theta} - \frac{\partial z_{\xi,\bar{y}}}{\partial \theta}.$$

We therefore have:

$$\nabla_\theta \mathcal{J}(\theta; \xi) = p(1-p)\, \mathbf{d}(\xi,\theta).$$

Taking norms yields:

$$\|\nabla_\theta \mathcal{J}(\theta; \xi)\| = p(1-p)\, \|\mathbf{d}(\xi,\theta)\|.$$

For fixed state $\xi$ and parameters $\theta$, the magnitude of the policy gradient is directly proportional to $p(1-p)$, with proportionality constant $\|\mathbf{d}(\xi,\theta)\| > 0$, which completes the proof. $\square$

## E   THE MEANS AND STANDARD DEVIATIONS OF MAIN RESULTS

We conduct multiple rounds of evaluation, and report the means and standard deviations of the main results in Table 7, and 8.

Table 7: Comparison of data selection methods on GSM8K test set. We train Qwen2.5-1.5B-Instruct on the GSM8K training selected subset. The mean and standard deviation of results are reported.

| Data Selection Method | Selected Data Size | | | |
|---|---|---|---|---|
| | 100 | 500 | 1,000 | 2,000 |
| Qwen2.5-1.5B-Instruct | 55.7±0.8 | | | |
| Qwen2.5-1.5B-Instruct-FULL | 77.0±0.3 | | | |
| Random Sampling | 74.0±0.7 | 74.8±0.5 | 74.9±0.1 | 75.8±0.2 |
| PPL-Top (Laurençon et al., 2022) | 72.2±0.3 | 73.1±0.1 | 73.8±0.6 | 75.3±0.4 |
| PPL-Middle (Ankner et al., 2024) | 73.9±0.1 | 74.8±0.5 | 74.8±0.4 | 75.3±0.6 |
| IFD (Li et al., 2023a) | 74.1±0.7 | 76.0±0.3 | 75.5±0.4 | 76.1±0.5 |
| Token Length (Xia et al., 2024b) | 74.0±0.3 | 75.4±0.5 | 75.3±0.4 | 76.3±0.7 |
| SelectIT (Liu et al., 2024a) | 74.2±0.3 | 75.0±0.2 | 75.4±0.6 | 75.0±0.4 |
| LIMR (Li et al., 2025) | 74.2±0.3 | 75.6±0.7 | 75.5±0.6 | 76.3±0.3 |
| *LearnAlign* | **74.5±0.3** | **76.8±0.5** | **77.5±0.4** | **78.0±0.3** |

## F   THE SIGNIFICANCE OF "WARMUP TRAINING" AND "DATA LEARNABILITY"

To fully show the role of "warmup training" and "data learnability", we conducted the ablation experiments on Qwen2.5-3B and Qwen2.5-7B by training for 2,000 and 1,000 steps, respectively. As shown in Table 9, both of them have a significant impact on the proposed method through sufficient training.

## G   DETAILED DISCUSSION ON PRACTICAL DATA SELECTION TIME COST AND COMPUTATIONAL EFFICIENCY

This appendix provides a detailed discussion on the computational efficiency of LearnAlign-based data selection. We elaborate on (1) efficient implementation of gradient-information estimation (Step 2), (2) the efficiency of LearnAlign score computation despite the nominal $n \times n$ matrix size (Step 4), and (3) a comparison of time costs against baseline methods.

Table 8: Comparison of data selection methods on four math benchmarks (GSM8K, MATH500, AMC2023, AIME2024) and one code benchmark (CRUX). We train Qwen2.5-3B and Qwen2.5-7B on the DAPO-MATH-17K selected subset with 1,000 data points. The mean and standard deviation of results are reported.

| Data Selection Method | GSM8K | MATH500 | AMC2023 | AIME2024 | CRUX |
|---|---|---|---|---|---|
| Qwen2.5-3B | 20.3±1.3 | 52.9±1.0 | 9.2±1.5 | 3.3±0.0 | 15.2±0.6 |
| Qwen2.5-3B-FULL | 82.5±0.5 | 64.3±0.4 | 32.5±0.2 | 20.0±5.4 | 22.8±0.3 |
| Random Sampling | 73.4±0.4 | 57.3±0.9 | 24.7±0.9 | 12.2±1.5 | 17.4±1.1 |
| PPL-Top (Laurençon et al., 2022) | 74.0±1.2 | 58.0±2.2 | 25.2±0.3 | 13.3±4.7 | 19.1±0.7 |
| PPL-Middle (Ankner et al., 2024) | 72.2±0.2 | 52.7±0.5 | 23.5±0.5 | 14.4±4.2 | 18.9±1.6 |
| IFD (Li et al., 2023a) | 69.6±0.4 | 56.0±1.8 | 22.7±0.4 | 15.5±6.8 | 17.9±0.2 |
| Token Length (Xia et al., 2024b) | 63.3±0.8 | 52.7±1.3 | 20.4±0.1 | 13.3±5.4 | 18.1±0.2 |
| SelectIT (Liu et al., 2024a) | 68.8±0.7 | 55.1±0.9 | 23.1±1.6 | 13.3±0.0 | 14.7±0.4 |
| LIMR (Li et al., 2025) | 73.6±0.8 | 57.3±1.5 | 23.3±0.4 | 15.6±3.1 | 17.1±0.3 |
| *LearnAlign* | **79.3±0.6** | **61.1±1.1** | **29.3±1.4** | **16.7±3.2** | **21.1±0.9** |
| Qwen2.5-7B | 27.0±1.1 | 66.1±1.5 | 17.9±0.3 | 17.8±1.9 | 25.7±0.8 |
| Qwen2.5-7B-FULL | 89.8±0.1 | 73.9±0.5 | 49.2±0.7 | 32.2±1.6 | 52.4±1.1 |
| Random Sampling | 82.9±0.5 | 64.5±0.5 | 30.2±0.5 | 23.3±0.0 | 41.0±0.5 |
| PPL-Top (Laurençon et al., 2022) | 83.6±0.8 | 64.4±0.4 | 27.4±0.4 | 27.8±3.1 | 44.0±1.5 |
| PPL-Middle (Ankner et al., 2024) | 79.5±1.2 | 64.6±0.5 | 27.8±0.1 | 17.8±3.1 | 38.9±0.6 |
| IFD (Li et al., 2023a) | 82.8±1.1 | 63.1±0.4 | 27.6±1.0 | 20.0±0.0 | 35.3±0.4 |
| Token Length (Xia et al., 2024b) | 78.7±0.9 | 62.1±2.1 | 25.8±0.4 | 23.3±4.7 | 34.4±2.9 |
| SelectIT (Liu et al., 2024a) | 84.6±0.2 | 64.3±1.5 | 27.7±0.1 | 18.9±3.1 | 40.8±0.8 |
| LIMR (Li et al., 2025) | 82.7±1.3 | 65.7±1.6 | 28.0±0.4 | 25.6±3.1 | 39.4±0.9 |
| *LearnAlign* | **87.7±0.7** | **71.0±0.4** | **34.0±1.4** | **28.9±3.1** | **43.1±0.2** |

Table 9: Ablation study of warmup training and data learnability. We train Qwen2.5-3B and Qwen2.5-7B on the DAPO-MATH-17K selected 1,000 examples for 2,000 and 1,000 steps, respectively.

| Benchmark | GSM8K | MATH500 | AMC2023 |
|---|---|---|---|
| *LearnAlign*(Qwen2.5-3B, 2,000 steps) | 83.8±1.0 | 67.8±2.1 | 36.9 ±0.7 |
| w/o warmup training (Qwen2.5-3B, 2,000 steps) | 81.9±0.2 | 64.4±0.6 | 31.0±2.0 |
| w/o data learnability (Qwen2.5-3B, 2,000 steps) | 81.3±1.1 | 63.6±2.2 | 34.8 ±0.8 |
| *LearnAlign*(Qwen2.5-7B, 1,000 steps) | 90.4±0.4 | 76.7±0.4 | 48.6±0.3 |
| w/o warmup training (Qwen2.5-7B, 1,000 steps) | 89.9±0.1 | 73.2±0.2 | 43.7±0.3 |
| w/o data learnability (Qwen2.5-7B, 1,000 steps) | 89.9±0.3 | 75.3±0.5 | 46.4±0.7 |

## G.1 EFFICIENT IMPLEMENTATION OF GRADIENT INFORMATION ESTIMATION (STEP 2)

**Current efficiency measures in our method.** As shown in Table 10, the Gradient Information Estimation step (Step 2) is the most time-consuming part of our method. We adopt two strategies to make gradient-information estimation efficient:

- **Single-rollout gradient computation.** Following (Lin et al., 2025), we compute the gradient of a single correct rollout per sample, which significantly reduces backpropagation cost. Table 4 of the main paper shows that this yields substantial savings while preserving the informative gradient directions required for LearnAlign.

- **Random projection of gradients.** Full-dimensional gradients are prohibitively large. We adopt a Johnson–Lindenstrauss–style (Johnson et al., 1984) random projection:

$$\phi(\theta; x) = \Gamma^\top \nabla \mathcal{J}_{\text{GRPO}}(\theta; x), \qquad \Gamma \in \mathbb{R}^{d \times k}, \ k \ll d.$$

Table 10: Time cost of different steps in LearnAlign.

| Step | Time |
|---|---|
| Step 1: Warmup Training | 2h 2min |
| Step 2: Gradient Information Estimation | 4h 12min |
| Step 3: Learnability Computation | 2h 41min |
| Step 4: LearnAlign-based Data Selection | <1 min (12.7s) |
| **Total** | 8h 55min |

This preserves inner products, enabling efficient computation of LearnAlign scores in a low-dimensional space.

**Other possible techniques for efficient computation.**

- **Cancellation effect.** Prior work (Yeh et al., 2022) shows that token-level gradients can exhibit cancellation across time steps, allowing partial reuse of intermediate results and reducing redundant backpropagation.
- **LoRA-space gradients.** Instead of backpropagating through the full parameter space, one may compute gradients only within a low-rank LoRA subspace (Hu et al., 2022), dramatically reducing dimensionality while preserving informative update directions.
- **Neural-network surrogate models for influence prediction.** A potential direction is to train a compact neural network to predict influence scores from cheaper metadata (e.g., embeddings, rollout statistics). Prior studies (Agarwal & Hakkani-Tür) show such surrogate models can remove the need to compute full gradients for every sample.

### G.2 EFFICIENT IMPLEMENTATION OF LEARNALIGN SCORE MATRIX (STEP 4)

Although Step 4 conceptually involves an $n \times n$ LearnAlign score matrix, the computation is extremely efficient.

**Current data scales ($n = 10^3$–$10^4$).** In our experiments, the training set size is at most a few tens of thousands. Step 4 is implemented as a single batched GPU matrix multiplication on low-dimensional gradient features. Table 10 shows that LearnAlign selection takes only **12.7 seconds**, compared with over **4 hours** for gradient computation. The computational bottleneck lies overwhelmingly in obtaining gradients, not in matrix operations.

**Scalable extensions for ultra-large datasets.** When $n$ reaches hundreds of thousands, the following scalable methods can be further applied:

- **Low-rank/Nyström sampling.** Approximate the full similarity matrix using a small subset of rows/columns (e.g., via Nyström sampling (Williams & Seeger, 2000)), reducing cost from $O(n^2)$ to $O(nc)$, where $c$ is the number of sampled rows/columns and $c \ll n$.
- **Two-stage cascade selection** (Gong et al., 2025): Use a cheap embedding-based filter to reduce the candidate pool, then apply LearnAlign only on that smaller set.

### G.3 COMPUTATIONAL COST OF BASELINE METHODS

Table 11 summarizes the time cost of different data-selection baselines for training Qwen2.5-3B on DAPO-MATH-17K. All baselines include the same warmup training time (2h2min) and rollout sampling (2h41min).

## H DISCUSSION ON REPRESENTATIVENESS AND DIVERSITY

To assess the trade-off between diversity and representativeness, we conduct additional experiments that incorporate feature-space diversity (Xia et al., 2024b). For example, we combine K-means

Table 11: Comparison of data selection time cost for different methods.

| Method | Time |
|---|---|
| PPL-Top | 5h 34min |
| PPL-Middle | 5h 34min |
| IFD | 6h 26min |
| Token Length | 4h 44min |
| SelectIT | 6h 54min |
| LIMR | 43h 12min |
| *LearnAlign* | 8h 55min |

clustering with *LearnAlign*, selecting the highest-scoring samples within each cluster to promote diversity. As shown in Table 12, incorporating explicit feature-space diversity does not yield significant gains over *LearnAlign*, which prioritizes representativeness. Moreover, the diversity-aware variant remains sensitive to the choice of the number of clusters $k$.

As reported in LIMO (Ye et al., 2025), the reasoning capability stimulated by an example is not directly correlated with shallow features, making traditional diversity criteria (e.g., k-means over embeddings) unreliable. For RLVR reasoning, recent studies (Ye et al., 2025; Li et al., 2025) show that very small subsets of high-value reasoning data, even a one-shot example, can provide broad generalization improvements across categories. Overall, current evidence suggests that representative and learnable samples are the primary bottleneck for policy improvement, and feature-level diversity provides limited additional benefit. Therefore, our method prioritizes representativeness. Nevertheless, we acknowledge that diversity-aware RLVR data selection remains underexplored, and investigating principled diversity metrics beyond surface features is an important direction for future work.

Table 12: The performance of LearnAlign that integrates the K-means clustering on DAPO-MATH-14K with Qwen2.5-3B.

| Model | GSM8K | MATH500 | AMC2023 |
|---|---|---|---|
| *LearnAlign* | 79.3 | 60.2 | 28.3 |
| +k-means ($k$=5) | 77.5 | 59.8 | 27.1 |
| +k-means ($k$=10) | 80.3 | 60.8 | 27.4 |
| +k-means ($k$=20) | 78.4 | 60.4 | 26.8 |

## I    SENSITIVITY OF THE WARMUP DATASET

The warmup dataset also may affect the performance of *LearnAlign*. We perform experiments with three different warmup datasets on DAPO-MATH-14K with Qwen2.5-3B. As shown in Table 13, the proposed data selection method is robust to the randomness of the initial warmup dataset.

Table 13: The performance of LearnAlign with three different warmup datasets on DAPO-MATH-14K with Qwen2.5-3B.

| Model | GSM8K | MATH500 | AMC2023 |
|---|---|---|---|
| *LearnAlign*(warmup dataset 1) | 79.3 | 60.2 | 28.3 |
| *LearnAlign*(warmup dataset 2) | 79.5 | 61.8 | 29.5 |
| *LearnAlign*(warmup dataset 3) | 81.2 | 60.4 | 28.9 |

## J    COMPARISON WITH FILTERING DATA BY THE PASS@N SCORE

In addition to the existing baselines, we also consider a baseline that selects data using the pass@N score, which measures how often a model successfully solves a problem across N independent

Table 14: Comparison of data selection methods on three benchmarks. We train Qwen2.5-3B on the DAPO-MATH-17K with a selected subset.

| Data Selection Method | GSM8K | MATH500 | AMC2023 |
|---|---|---|---|
| Qwen2.5-3B-FULL (2,174 steps) | 83.6 | 65.8 | 31.0 |
| Learnability (2,000 steps) | 82.9 | 65.0 | 33.4 |
| Pass@8 Score Filter (2,000 steps) | 83.5 | 64.6 | 31.7 |
| *LearnAlign*(2,000 steps) | 83.8 | 67.8 | 36.9 |
| Qwen2.5-7B-FULL (1,000 steps) | 90.0 | 77.6 | 47.3 |
| Learnability (1,000 steps) | 89.9 | 74.4 | 46.4 |
| Pass@8 Score Filter (1,000 steps) | 89.9 | 75.0 | 43.7 |
| *LearnAlign*(1,000 steps) | 90.4 | 76.7 | 48.6 |

attempts. Specifically, we implement a pass@8–based filtering strategy: we remove questions whose pass@8 score falls in $\{0, 1, 7, 8\}$, as these correspond to samples that are either extremely easy or extremely difficult. Since the number of samples selected by pass@8–based filtering is not fixed, we train all Qwen2.5-3B and Qwen2.5-7B for about 2,000 steps and 1,000 steps respectively, to ensure a fair comparison. For completeness, we also include a learnability-only baseline that selects the 1,000 samples with the highest success-rate-based learnability introduced in this paper.

The comparison is shown in Table 14. Furthermore, we also highlight two key observations based on the actual results:

1. Learnability and pass@8 filtering achieve comparable performance by selecting medium-difficulty samples. Both methods aim to avoid overly easy and overly hard questions, and therefore select samples near the "middle" of the model's current capability. This results in comparable performance between the two methods across three benchmarks. Importantly, both methods achieve accuracy relatively close to full-data training, confirming the intuition that medium-difficulty samples carry substantial training value under RLVR.

2. LearnAlign outperforms full-data training and clearly surpasses baselines. In contrast to purely difficulty-based filtering, LearnAlign incorporates gradient-direction alignment to additionally capture the representativeness of each sample. As shown in Table 14, LearnAlign exceeds the full-data baseline and shows a clear margin over both Learnability and Pass@8 filtering across all three benchmarks. This demonstrates that combining learnability with gradient alignment signals yields a substantially more informative subset than using difficulty signal alone.

Overall, the results indicate that while selecting medium-difficulty samples is beneficial, considering gradient alignment is essential for identifying the truly most impactful RLVR data, leading to stronger and more consistent gains. Besides, we are running additional experiments and will update more results once we finish them.

## K  STAGED REINFORCEMENT LEARNING WITH LEARNALIGN

To further assess the applicability of our method in a curriculum learning scenario, we design a three-stage training procedure on the GSM8K dataset. Specifically:

- In the **first stage**, we use Qwen2.5-1.5B-Instruct to select the top 50% of the training samples, and train the model until convergence.

- In the **second stage**, the resulting model is used to select the next 30% of the samples, and again trained to convergence.

- In the **final stage**, the latest model selects the 20% of samples, and training is repeated until convergence.

As shown in Figure 5, our method can seamlessly integrate into such a staged RL curriculum to significantly improve capability acquisition.

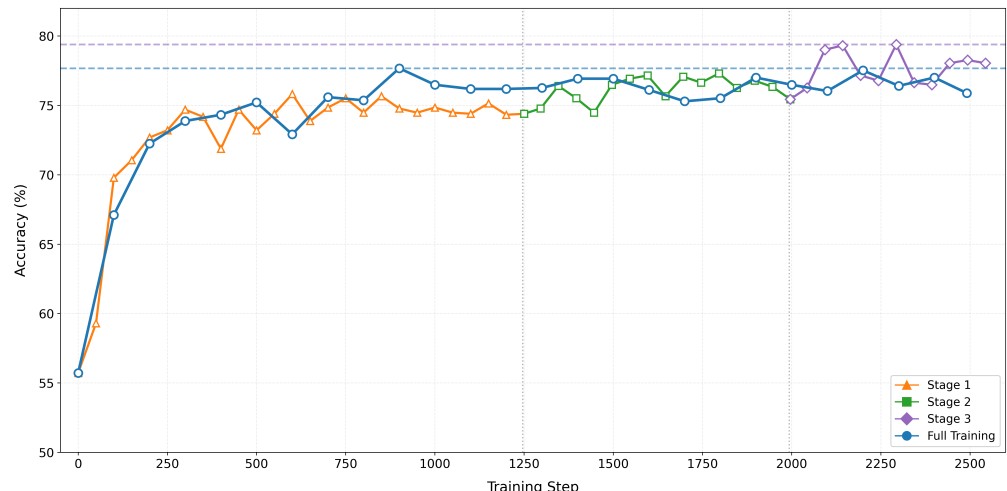

Figure 5: The performance of the staged reinforcement learning with the proposed data selection method.

## L  STATEMENT ON THE USE OF LARGE LANGUAGE MODELS

In the preparation period, we employ the large language model ChatGPT-5, developed by OpenAI, as a tool for writing assistance. Its role was strictly confined to enhancing language quality, including improvements in grammar, spelling, clarity, and sentence structure. The model was not utilized for generating scientific concepts, performing analyses, or interpreting results. All text produced by the model was thoroughly reviewed and edited by the authors, who assume full responsibility for the final content of this paper.

