# OpenReview forum: "Data Selection for LLM Reinforcement Learning with Improved Gradient Alignment"
_ICLR.cc/2026/Conference — ICLR 2026 Conference Withdrawn Submission_

### Official Review · Reviewer_bKxL · 2025-10-25

**Soundness:** 2
**Presentation:** 3
**Contribution:** 1
**Rating:** 2
**Confidence:** 4

**Summary:**

This paper introduces LearnAlign, a data selection method that combines gradient information and successful rate. The experiments show that using only 1,000 samples selected by this method can achieve comparable performance than training on the full dataset, and can outperform other data selection baselines on multiple test sets.

**Strengths:**

1. The idea of using gradient information to select representative data has certain theoretical significance.
2. The dataset obtained through this method is greatly reduced in size compared to the original dataset while achieving comparable performance.
3. The paper provides insights of how traditional SFT data selection methods perform in RLVR.

**Weaknesses:**

1. Lack of a clear comparison of convergence speed. Simply comparing results at the level of epochs may not be sufficient. Table 2 shows that the full dataset achieves a better final result. A natural question is: Under the same number of training steps, how does the performance of the full dataset compare to LearnAlign?  A curve of val_score versus training step would help visualize and compare the convergence speed on the full dataset and on the selected dataset.

2. Lack of discussion of training convergence behavior and the final performance ceiling. In general, a larger dataset can support longer training and provide a better final result. Can the selected data, by training for more epochs, eventually reach the same performance level as the full dataset?

3. The method relies on a warm-up training phase to obtain gradient information. However, in RLVR the model’s output distribution typically shifts substantially during training, so the initial gradient information may not reliably indicate which samples are useful. Moreover, the warm-up phase requires fully processing the entire dataset and computing the LearnAligner score, which has O(n²) complexity, leading to significant computational overhead.

4. Missing key baseline. A very common practice in RLVR is to use the pass@N score to filter data. For example, one can drop questions whose pass@8 score is 0/1/7/8, because such questions are either too easy or too hard. Since this approach is extremely easy to implement and is already used as a filtering step before many RLVR training pipelines, the authors should consider comparing against this basic method.

**Questions:**

Please consider addressing the issues raised under Weaknesses.

---

> ### Author Response · Authors · 2025-11-26
>
> Thanks for your constructive comments. We address your concerns as follows:
>
> > Q1:There is no clear comparison of convergence speed. Under the same number of training steps, how does the full dataset perform compared with LearnAlign? A val-score–vs.–step curve would help illustrate the difference.
>
> **A1:** We appreciate the reviewer’s helpful suggestion to compare convergence under the same number of optimization steps. We train Qwen2.5-3B with 16 epochs (2000 steps) to compare with the FULL-data training (2174 steps).
>
> As shown in the updated validation-accuracy-versus-training-step curve in **Section 5.3**, LearnAlign consistently maintains higher validation accuracy throughout almost the entire training trajectory when compared step-by-step with FULL training. This indicates that the selected data provide more effective learning signals per gradient update.
>
> A particularly clear comparison arises when examining the steps required to reach the FULL performance peak. The FULL-data training reaches its highest accuracy of **63.12\%** at **step 640**. In contrast, LearnAlign reaches the same accuracy at only **step 440**, requiring **31\% fewer training steps** to achieve the same performance level. This demonstrates that LearnAlign achieves substantially faster convergence under identical training budgets, providing strong evidence for the high training efficiency of the proposed method.
>
> > Q2: Lack of discussion on convergence behavior and performance limits. Can the selected subset, given more epochs, eventually reach the full-dataset performance?
>
> **A2:** We thank the reviewer for raising the important question regarding convergence behavior and performance limits. To examine whether the selected subset constrains the final achievable performance, we trained the LearnAlign-selected data for additional epochs (4, 8, and 16). As shown in Table 4-1, extended training on the selected subset not only reaches the FULL-dataset performance but in fact surpasses it.
>
> As illustrated in the updated training curve in **Section 5.3**, LearnAlign continues improving after matching the FULL peak and eventually attains a higher maximum accuracy of 64.22\% at step 1040. After reaching this peak, the LearnAlign curve remains stable and exhibits a smoother late-stage plateau compared to FULL-data training, indicating that the selected data do not cause premature saturation or overfitting.
>
> These findings confirm that the selected subset does not impose a performance ceiling; rather, it retains sufficient learnability to sustain continued improvements when trained for more epochs. This observation aligns with prior works [1, 2], which suggest that precise sample selection, rather than the raw quantity of data, plays a central role in unlocking enhanced reasoning capabilities during RL post-training.
>
> Table 4-1.  Comparison of data selection methods. We train Qwen2.5-3B on the DAPO-MATH-17K selected subset with 1,000 data points.
> | Data Selection Method | GSM8K | MATH500 | AMC2023 |
> | --------------------- | ----- | ------- | ------- |
> | Qwen2.5-3B-FULL(2174 steps, 1 epochs)       |   83.6     | 65.8        |    31.0     |
> | LearnAlign (250 steps, 2 epochs)  |   79.3    | 60.2        |    28.3     |
> | LearnAlign (500 steps, 4 epochs)  |    80.7   |  63.4      |   31.5 |
> | LearnAlign (1000 steps, 8 epochs)  |  82.9 | 64.6|   35.2     |
> | LearnAlign (2000 steps, 16 epochs) |  83.8   |       67.8 |    36.9 |
>
> [1] LIMR: Less is More for RL Scaling. arXiv 2025
>
> [2] Reinforcement Learning for Reasoning in Large Language Models with One Training Example. NeurIPS 2025
>
> > Q3: The method relies on warmup gradients, which may be unreliable due to RLVR distribution shifts, and computing the O(n²) LearnAlign scores creates significant overhead.
>
>
> **A3:** Thank you for raising this concern. We explain it as follows:
>
> **1. The warmup gradients are informative.** Our use of a warmup phase is not based on the claim that early gradients remain exactly unchanged but on the more modest and empirically supported observation that early training dynamics already contain informative signals about useful directions in parameter space. Prior works have shown that early gradients strongly shape later optimization trajectories [3], and the early curvature/gradient statistics correlate with final generalization [4,5].
>
> In addition, Appendix K in the paper shows that multi-stage LearnAlign may yield further improvements when more computing resources are available, which illustrates an efficiency–performance trade-off for applying our method.
>
> [3] The Early Phase of Neural Network Training. ICLR 2020
>
> [4] The Role of the Loss Landscape Geometry in the Generalization of Deep Networks. ICLR 2021
>
> [5] Drawing Early-Bird Tickets: Towards More Efficient Training of Deep Networks. ICLR 2020

---

> ### Author Response · Authors · 2025-11-26
>
> **2. The proposed method is relatively computationally efficient.**
> Warmup training does not require processing the entire dataset. In our method, it is performed only on a small randomly sampled subset, as its purpose is merely to “activate’’ the model and capture representative early training signals. Thus, the complexity of warmup training depends on the size of warmup dataset $m$, that is much smaller than $n$. Besides, as shown in Table 2-1, the computation of LearnAlign score takes only 12.7 seconds, due to the efficient CUDA-optimized matrix multiplication and the low-dimensional projection of gradients. For scalability considerations of the computation, please refer to **A2** for Reviewer JWPt.
>
> > Q4: A key baseline is missing: filtering data via the pass@N score, a widely used and trivial RLVR practice. The method should be compared against this basic baseline.
>
> **A4:** Thank you for pointing out this important baseline. As suggested, we implemented the commonly used pass@8-score filtering, which removes questions whose pass@8 score falls in {0,1,7,8}, corresponding to extremely easy or extremely hard samples. Since the number of samples selected by pass@8–based filtering is not fixed, we train all Qwen2.5-3B and Qwen2.5-7B for about 2,000 steps and 1,000 steps, respectively, to ensure a fair comparison. For completeness, we also include a learnability-only baseline that selects the 1000 samples with the highest success-rate-based learnability $p(1−p)$ introduced in our paper.
>
> The comparison is shown in Table 4-2. We highlight two key observations based on the actual results:
>
> **1. Learnability and pass@8 filtering achieve comparable performance by selecting medium-difficulty samples.** Both methods aim to avoid overly easy and overly hard questions, and therefore select samples near the “middle” of the model’s current capability. This results in comparable performance between the two methods across three benchmarks. Importantly, both methods achieve accuracy relatively close to full-data training, confirming the intuition that medium-difficulty samples carry substantial training value under RLVR.
>
> **2. LearnAlign outperforms full-data training and clearly surpasses baselines.** In contrast to purely difficulty-based filtering, LearnAlign incorporates gradient-direction alignment to additionally capture the representativeness of each sample. As shown in Table 4-2, LearnAlign exceeds the full-data baseline and shows a clear margin over both Learnability and Pass@8 filtering across all three benchmarks. This demonstrates that combining learnability with gradient alignment signals yields a substantially more informative subset than using difficulty signal alone.
>
> Overall, the results indicate that while selecting medium-difficulty samples is beneficial, considering gradient alignment is essential for identifying the truly most impactful RLVR data, leading to stronger and more consistent gains.
>
> Table 4-2. Comparison of data selection methods on three benchmarks. We train Qwen2.5-3B on the DAPO-MATH-17K with a selected subset.
> | Data Selection Method | GSM8K | MATH500 | AMC2023 |
> | --------------------- | ----- | ------- | ------- |
> | Qwen2.5-3B-FULL (2174 steps)       |   83.6     | 65.8        |    31.0     |
> |  Learnability (Qwen2.5-3B, 2000 steps)   | 82.9      |  65.0      |    33.4     |
> | Pass@8 Score Filter (Qwen2.5-3B, 2000 steps)  | 83.5      |  64.6      |    31.7     |
> | **LearnAlign (Qwen2.5-3B, 2000 steps)** | 83.8 | 67.8 | 36.9 |
> | Qwen2.5-7B-FULL (1000 steps)       |   90.0    | 77.6      |    47.3     |
> |  Learnability (Qwen2.5-7B, 1000 steps)   | 89.9    |  74.4     |   46.4     |
> | Pass@8 Score Filter (Qwen2.5-7B, 1000 steps)  | 89.9     | 75.0     |    43.7   |
> | **LearnAlign (Qwen2.5-7B, 1000 steps)** | 90.4 | 76.7 | 48.6 |

---

> ### Author Response · Authors · 2025-12-03
>
> **Dear Reviewer bKxL,**
>
> Thank you very much for your constructive and insightful comments. As the rebuttal period concludes, we would like to briefly summarize the revisions and additional experiments we conducted to address your concerns. All corresponding updates have been incorporated into the revised manuscript, including new analyses, figures, and tables.
>
> In response to your questions, we have made the following revisions and additions, and they have been incorporated into the updated manuscript and involve substantial additional experiments and figures:
>
> **Added convergence-speed comparison with new training curves.**
> Following your suggestion, we trained LearnAlign and FULL under matched optimization steps and added a new validation-accuracy–versus–training-step curve (Section 5.3). This figure clearly shows that LearnAlign achieves the FULL peak accuracy 31% faster, demonstrating substantially improved convergence efficiency.
>
> **Extended convergence-limit experiments with new tables.**
> We conducted additional multi-epoch training (4, 8, 16 epochs), and the new results are now reported in Table 4-1. These experiments show that LearnAlign not only reaches the FULL-dataset performance but eventually surpasses it, providing a clearer understanding of long-horizon convergence behavior.
>
> **Clarified the warmup reliability and reduced-complexity design.**
> In response to your concerns about warmup gradients and O(n²) computation, we expanded the discussion in **Section 5.4** and **Appendix G**. We highlight that warmup is applied to a small sampled subset, and Step 4 uses CUDA-accelerated low-dimensional matrix operations, taking only seconds. These clarifications are now reflected in the revised manuscript.
>
> **Added a new baseline comparison with pass@N filtering.**
> As suggested, we implemented the widely used pass@8 filtering and included the results in Table 4-2. This new table shows that while difficulty-only filtering performs reasonably well, LearnAlign achieves clearly superior performance, reinforcing the importance of combining gradient representativeness with learnability.
>
> Together, these updates introduce new figures, extended analyses, and multiple new experimental results, strengthening the empirical evaluation and clarifying the convergence and scalability characteristics of LearnAlign.
>
> We hope these revisions have fully addressed your concerns.We sincerely appreciate your thoughtful feedback and the opportunity to improve our research.
>
> Best regards,
>
> Authors

---

### Official Review · Reviewer_HRzu · 2025-10-28

**Soundness:** 3
**Presentation:** 3
**Contribution:** 3
**Rating:** 6
**Confidence:** 3

**Summary:**

This paper addresses the significant data inefficiency of Reinforcement Learning with Verifiable Rewards (RLVR) used for enhancing LLM reasoning. The authors propose a novel data selection method called LearnAlign, which selects a small, high-quality subset of learnable and representative data for training. Experiments across reasoning benchmarks show that LearnAlign significantly outperforms existing data selection methods.

**Strengths:**

* The paper tackles a practical and important bottleneck in post-training LLMs: the high cost and data inefficiency of RLVR.
* The paper is well-written and easy to follow.
* The data selection process defined in the paper is quite reasonable and logically structured.

**Weaknesses:**

* The proposed data learnability takes the form of a simple quadratic function. While using the success rate (p) as a basis for the metric is logical, it is questionable whether the metric had to be in this specific multiplicative form. The method would be more persuasive if sufficient justification or additional experiments (e.g., exploring alternative functions) were provided for this design choice.
* Since the entire data selection process is determined by the warm-up model, there is a concern that it might be overly dependent on the specific random dataset chosen for this warm-up. Although the authors showed in Table 3 that warm-up training is a necessary component, it is unclear how sensitive the selection process is to the randomness of this initial dataset.

**Questions:**

* This paper seems to propose a method for learning well by selecting representative data from the dataset. How does this focus on representativeness affect the learning process in terms of diversity? What are the authors' opinions on the trade-off and balance between selecting representative data versus diverse data?
* Regarding the results in Table 4, what are the computational costs for the other baselines, such as the PPL-based methods or SelectIT?

---

> ### Author Response · Authors · 2025-11-26
>
> Thanks a lot for your kind comments. We address your concerns as follows:
>
> > Q1: The justification for the data learnability form of a simple quadratic function.
>
> **A1:** We thank the reviewer for the insightful question. Although the proposed learnability metric $p(1−p)$ may appear simple, it is in fact a theoretically grounded formulation for modeling learnability under Bernoulli feedback in RLVR.
>
> First, the success rate $p$ measures how often the model receives informative positive trajectories revealing correct behavior, while $1−p$ captures the remaining room for improvement. A sample provides a useful learning signal only when both conditions co-exist, and thus their product $p(1−p)$ can represent the expected improvement that a data point will provide [1,2].
>
> Second, $p(1−p)$ is precisely the variance of Bernoulli accuracy rewards. Recent theoretical analyses [3,4] show that the reward variance lower-bounds the KL divergence between the initial and the optimal model, making it an effective statistical quantity reflecting the gradient informativeness of a sample.
>
> Third, this quadratic form is not arbitrary: it is the unique smooth, symmetric, unimodal function that (i) peaks at intermediate difficulty, (ii) vanishes at $p=1$, and (iii) aligns with Fisher-information–based measures of sample utility [5]. Alternative function choices fail to satisfy these properties or lack comparable theoretical interpretability.
>
> Last but not least, as analyzed in **Appendix D** of our updated paper, for a fixed query $\xi\$ and model $\theta\$, the policy gradient magnitude is positively proportional to $p(1-p)$. It indicates that $p(1-p)$ can represent the information about the gradient magnitude without the issue of response-length bias.
>
> Overall, for these reasons, $p(1−p)$ is a principled, theoretically grounded, and empirically supported metric for modeling learnability.
>
> [1] Automatic Goal Generation for Reinforcement Learning Agents. ICML 2018
>
> [2] Proximal Curriculum for Reinforcement Learning Agents. TMLR 2023
>
> [3] What Makes a Reward Model a Good Teacher? An Optimization Perspective. arXiv 2025
>
> [4] Online Difficulty Filtering for Reasoning-Oriented Reinforcement Learning. arXiv 2025
>
> [5] Information-based objective functions for active data selection. Neural Computation 1992
>
>
>
> > Q2:How sensitive the selection process is to the randomness of the initial dataset?
>
> **A2:** Thanks a lot for your insightful comment. We have conducted the experiments with three different warmup datasets on DAPO-MATH-14K with Qwen2.5-3B. As shown in Table 3-1，the proposed data selection method is robust to the randomness of the initial warmup dataset.
>
> Table 3-1. The performance of LearnAlign with three different warmup datasets on DAPO-MATH-14K with Qwen2.5-3B.
> | Model               | GSM8K | MATH500 | AMC2023 |
> | --- | --- | -- | - |
> | LearnALign (warmup dataset 1) |79.3    | 60.2        |    28.3     |
> | LearnALign (warmup dataset 2) |79.5    |       61.8 |    29.5     |
> | LearnALign (warmup dataset 3) |   81.2   |       60.4   |    28.9     |
>
> > Q3: How does this focus on representativeness affect the learning process in terms of diversity? What are the authors' opinions on the trade-off and balance between selecting representative data versus diverse data?
>
> **A3:** Thank you for the insightful question. Our method intentionally emphasizes representativeness—i.e., samples that most effectively reduce the global training loss via high gradient alignment. We discuss its relationship to diversity below.
>
> **Representativeness as the key driver of RLVR data influence.**
> Our goal is to identify data that meaningfully influence model updates under RLVR. Recent studies [7,8] on reasoning LLMs show that very small subsets of high-value reasoning data, even one-shot example, can provide broad generalization improvements across categories. These results indicate that, for RLVR, representativeness is the main driver of capability acquisition. Empirically, our selected subsets achieve performance comparable to or even exceeding full-data training, supporting this view.
>
> **Does focusing on representativeness reduce diversity?**
> We agree this could, in principle, underrepresent rare but useful patterns. However, measuring “diversity” in RLVR reasoning data remains an open problem, because reasoning capability is not well predicted by superficial features such as prompt length, semantic clusters, or topics. As reported in LIMO [7], the reasoning capability stimulated by an example is not directly correlated with shallow features, making traditional diversity criteria (e.g., k-means over embeddings) unreliable.
>
> [7] LIMO: Less Is More for Reasoning. arXiv 2025
>
> [8] LIMR: Less is More for RL Scaling. arXiv 2025

---

> ### Author Response · Authors · 2025-11-26
>
> We have experimented with adding feature-space diversity [9], i.e., integrating the K-means clustering with the proposed method, that selects higher-scoring data within each cluster to boost the diversity. As shown in Table 3-2, adding such diversity on top of LearnAlign did not improve significant performance.
>
> **Our opinions.**
> For RLVR reasoning, current evidence suggests that representative and learnable samples are the primary bottleneck for policy improvement, and feature-level diversity provides limited additional benefit. Therefore, our method prioritizes representativeness. Nevertheless, we acknowledge that diversity-aware RLVR data selection remains underexplored, and investigating principled diversity metrics beyond surface features is an important direction for future work.
>
> Table 3-2. The performance of LearnAlign that integrates the K-means clustering on DAPO-MATH-14K with Qwen2.5-3B.
> | Model           | GSM8K | MATH500 | AMC2023 |
> | --------------- | ----- | ------- | ------- |
> | LearnAlign      | 79.3    | 60.2        |    28.3     |
> | + k-means (k=5) | 77.5 | 59.8   | 27.1   |
> | + k-means (k=10)   | 80.3   |  60.8       |  27.4       |
> | + k-means (k=20) | 78.4   | 60.4    | 26.8   |
>
> [9] Rethinking Data Selection at Scale: Random Selection is Almost All You Need. EMNLP Findings 2025
>
> > Q4:The computational costs for the other baselines.
>
> **A4:** Thanks for your kind comments. The computational costs of data selection for other baselines in training Qwen2.5-3B on DAPO-MATH-17K are shown in Table 3-3. Note that all baselines include the warmup training time (2h2min) and rollout sampling (2h41min).
>
> Table 3-3. Comparison of data selection time cost for different methods.
>
> | Step         | Time |
> | ------------ | ---- |
> | PPL-Top      | 5h34min |
> | PPL-Middle   | 5h34min |
> | IFD          | 6h26min  |
> | Token Length | 4h44min |
> | SelectIT     | 6h54min    |
> | LIMR         | 43h12min |
> | LearnAlign   | 8h55min |

---

> > ### Comment · Reviewer_HRzu · 2025-11-27
> >
> > Thank you for the comprehensive clarification.
> > After carefully considering your response, I will stand by my initial assessment.

---

> ### Author Response · Authors · 2025-11-27
>
> Thank the reviewer for the time and effort they have spent on our paper. We would also be happy to answer any further questions.

---

> ### Author Response · Authors · 2025-12-03
>
> **Dear Reviewer HRzu,**
>
> Thank you very much for your thoughtful and constructive comments. As the rebuttal period is coming to an end, we would like to kindly check whether our responses and the corresponding manuscript updates have sufficiently addressed your concerns.
>
> In response to your questions, we have made the following revisions and additions, many of which have been incorporated into the updated manuscript:
>
> **Justification of the learnability formulation.**
> We expanded the theoretical explanation of the $p(1-p)$ learnability metric in the Method section, detailing its grounding in Bernoulli-variance analysis, expected improvement theory, Fisher-information properties. These clarifications were added directly to the paper to strengthen conceptual justification.
>
> **New robustness experiments with additional tables.**
> To address sensitivity to warmup randomness, we conducted three independent runs using different warmup datasets. The new experimental results are presented in Table 3-1, now included in the appendix. These results demonstrate that LearnAlign remains stable across different initial warmup sets.
>
> **Expanded discussion and experiments on diversity.**
> We added a detailed discussion on the representativeness–diversity trade-off and conducted additional experiments integrating feature-space diversity using K-means clustering. The new results are presented in Table 3-2, showing that enforcing feature-level diversity does not consistently improve RLVR outcomes. This updated analysis is now included in the revised paper to reflect the empirical findings.
>
> **Detailed computational cost comparison across baselines.**
> In response to Question 4, we added a comprehensive comparison of computation time for all baselines, which now appears as Table 3-3. This table provides a clear view of the relative efficiency of LearnAlign compared to existing approaches.
>
> Together, these additions significantly strengthen the experimental evaluation and theoretical clarity of our work, and all new tables and analyses have been incorporated into the revised manuscript.
>
> We hope these revisions fully address your concerns. Thank you once again for your insightful comments and for helping us strengthen the paper.
>
> Best regards,
>
> Authors

---

### Official Review · Reviewer_JWPt · 2025-10-31

**Soundness:** 2
**Presentation:** 2
**Contribution:** 3
**Rating:** 6
**Confidence:** 3

**Summary:**

This work investigates data selection strategy for RLVR training based on gradient estimation for each training datapoint. To estimate the contribution of each training sample, a score for each pair of datapoints are computed that takes into account the pass rate of and the low rank estimation of the gradients on the datapoints. Finally, the top-N datapoints with the highest scores are selected as the training dataset. This approach successfully identifies valuable subsets that could achieve competitive performance as the full training set. Comparison with different data selection approach also shows that the proposed method outperforms existing approaches.

**Strengths:**

1. The paper is well written and description of the method is easy to follow.
2. The experiment section is comprehensive, covering performance comparison with baseline methods, ablation study, and time cost analysis.
3. Besides the time cost experiment, the paper also provides a theoretical analysis on the data collection cost.

**Weaknesses:**

See questions.

**Questions:**

1. While the proposed approach has shown to be effective according to the experiment results, is there any theoretical analysis on why the proposed approach work?
2. How could the gradient information estimation step be implemente efficiently?
3. In Step 4,the LearnAlign score matrix is extremely large since n is the size of training dataset. How is this step implemented efficiently?

---

> ### Author Response · Authors · 2025-11-25
>
> Thanks for your kind comments. We address your concerns as follows:
>
> > Q1: Is there any theoretical analysis explaining why it should work?
> >
> **A1:** Thanks a lot for your constructive question. We explain why the proposed approach works as follows:
>
> **1. Gradient alignment captures first-order optimization influence.** Following gradient-tracing–based influence estimation [1,2,3], the first-order Taylor expansion of the loss (see Eq. (2) in the paper) shows that the gradient inner product $∇𝐽(𝜉_𝑖)⋅∇𝐽(𝜉_𝑗)$ approximates the first-order effect of gradient updating with $𝜉_𝑖$ on the loss of $𝜉_𝑗$. Theoretically, samples whose gradients align well are expected to reduce the overall training loss. By averaging this alignment score over all other samples, the proposed method can select data that are globally representative from the optimization perspective.
>
> **2. Learnability is essential beyond gradient directional alignment.** Prior analyses show that gradient norms are biased by response length in LLMs [3,4], and gradient directional similarity alone does not indicate whether a data point is learnable for the current model. While, as highlighted in related RL studies [5-7], effective RL post-training requires alignment between task difficulty and the model’s capabilities.  Therefore, incorporating learnability is necessary to ensure selecting samples that the model can benefit from. To address this, we introduce a learnability weight $V(ξ)=p(1−p)$, which peaks at moderate difficulty and naturally emphasizes samples that the model can benefit from. As for the form of data learnability, please refer to **A1** for Reviewer HRzu. Note that we provide a theoretical justification of the form in **Appendix D**.
>
> Overall, this synergy between gradient directional alignment and learnability yields a subset that is both representative and learnable.
>
> [1] A Survey of Data Attribution: Methods, Applications, and Evaluation in the Era of Generative. AIJ 2025
>
> [2] Estimating training data influence by tracing gradient descent. NeurIPS 2020
>
> [3] LESS: Selecting influential data for targeted instruction tuning. ICML 2024
>
> [4] Understanding R1-zero-like training: A critical perspective. arXiv 2025
>
> [5] How difficulty-aware staged reinforcement learning enhances LLMs' reasoning capabilities: a preliminary experimental study. arxiv 2025
>
> [6] Automatic goal generation for reinforcement learning agents. ICML 2018
>
> [7] Proximal curriculum for reinforcement learning agents. arXiv 2023
>
> > Q2:How could the gradient information estimation step be implemented efficiently?
>
>
> **A2:** Thank you for the question. We clarify how gradient-information estimation in LearnAlign can be implemented efficiently, and outline additional extensions.
>
> **1. Current efficiency measures in our method:**
>
> We have already adopted two concrete efficiency strategies:
>
> **Single-rollout gradient computation.** Following [8], we compute the gradient of one correct rollout per sample under resource constraints. As shown in Table 4 of the paper, this significantly reduces backward-pass cost while preserving useful update directions.
>
> **Random projection of gradients.** Full-dimensional gradients are extremely costly to store and manipulate. We therefore employ Johnson–Lindenstrauss–style random projections [9], applying a fixed random projection matrix $\Gamma \in \mathbb{R}^{k \times d}$ with $k \ll d$ to each gradient vector. This yields a low-dimensional gradient representation $\phi(\theta; x) = \Gamma^\top \nabla\mathcal{J}_{\text{GRPO}}(\theta; x) \in \mathbb{R}^k.$ Such projections preserve inner products with high probability, enabling efficient approximation of LearnAlign scores while retaining the key geometric structure of gradients.
>
>
> **2. Other possible techniques for efficient computation:**
>
> **Cancellation effect.** Prior work [10] shows that token-level gradients can exhibit cancellation across time steps, allowing partial reuse of intermediate results and reducing redundant backpropagation.
>
> **LoRA-space gradient estimation.** Instead of backpropagating through the full parameter space, one may compute gradients only within a low-rank LoRA subspace [11], dramatically reducing dimensionality while preserving informative update directions.
>
> **Neural-network surrogate models for influence prediction.** A potential direction is to train a compact neural network to predict influence scores from cheaper metadata (e.g., embeddings, rollout statistics). Prior studies [12] show such surrogate models can remove the need to compute full gradients for every sample.
>
> [8] CPPO: Accelerating the training of group relative policy optimization-based reasoning models, NeurIPS 2025
>
> [9] Extensions of Lipschitz mappings into a Hilbert space. 1984
>
> [10] First is Better Than Last for Language Data Influence, NeurIPS 2022
>
> [11]  LoRA Without Regret, ThinkingMachines.ai. 2025
>
> [12] Gradient-Based Meta-Learning with Learned Surrogate Losses. ICML 2020

---

> ### Author Response · Authors · 2025-11-25
>
> > Q3:Since Step 4 requires an n×n LearnAlign score matrix, how is this step implemented efficiently?
>
> **A3:** In our experiments, as shown in Table 2-1, when training Qwen2.5-3B on the DAPO-MATH-17K, Step 4 (data selection based on LearnAlign) takes only 12.7 seconds, compared to 4.2 hours for gradient estimation (Step 2). The key reason is that we implement Step 4 as a single batched matrix operation on GPU: we first obtain low-dimensional gradient features via random projection and then compute the LearnAlign scores using the CUDA-accelerated matrix multiplication. For the current data scale ($10^3$ to $10^4$), this operation is extremely fast relative to the cost of computing gradients themselves.
>
> For ultra-large datasets, where the size of training data $n$ may be hundreds of thousands, the following scalable methods can be further applied:
>
> **Low-rank approximation via column/row sampling:** Approximate the full similarity matrix using a small subset of rows/columns (e.g., via Nyström sampling [14]), reducing cost from $O(n^2)$ to $O(nc)$, where $c$ is the number of sampled rows/columns and $𝑐≪𝑛$.
>
> **Two-stage cascade selection [15]:** Use a cheap embedding-based filter to reduce the candidate pool, then apply LearnAlign only on that smaller set.
>
> Table 2-1. Time cost of different steps in the proposed method.
> | Step                            | Time |
> | ------------------------------- | :--------: |
> | Step 1: Warmup Training      | 2h2min   |
> | Step 2: Gradient Information Estimation |  4h12min    |
> | Step 3: Learnability Computation        |   2h41min  |
> | Step 4: Data Selection based on LearnAlign  | <1min(12.7s)     |
> | Total  |   8h55min   |
>
> [14] Using the Nyström method to speed up kernel machines. NIPS 2000
>
> [15] A two-stage data selection framework for data-efficient model training on edge devices. KDD 2025

---

> ### Author Response · Authors · 2025-12-03
>
> **Dear Reviewer JWPt,**
>
> Thank you very much for your insightful comments and helpful suggestions. As the rebuttal period is approaching its end, we would like to kindly check whether our responses and the corresponding revisions have adequately addressed your concerns.
>
> In response to your questions, we have made the following updates, many of which have been incorporated into the revised manuscript:
>
> **Added theoretical clarification.**
> We expanded the method section to provide a clearer theoretical explanation of why LearnAlign works, including the first-order optimization view of gradient alignment and the necessity of learnability. These theoretical insights have been added to the paper to strengthen methodological grounding.
>
> **Added detailed explanation of efficient gradient-information estimation.**
> We added a new subsection in the appendix describing how gradient information can be estimated efficiently in practice. This includes single-rollout gradient computation (following recent work on constrained-cost RLHF optimization) and Johnson–Lindenstrauss–style random-projection–based gradient compression, which preserves inner products while reducing dimensionality. We further discuss additional potential extensions drawn from prior work, such as cancellation-aware gradient reuse, LoRA-subspace gradient estimation, and surrogate-model–based influence prediction.
>
> **Added scalability analysis and supporting table.**
> We included a detailed time-cost analysis of all steps (Table 2-1) in the appendix, showing that LearnAlign selection takes only seconds compared with hours of gradient estimation. We also expanded the discussion on scalable approximations (low-rank sampling, block-wise averaging).
>
> Overall, we have strengthened both the theoretical explanation and the practical implementation details while adding new tables and analyses that can be directly incorporated into the submission.
>
> We hope these revisions have fully addressed your concerns. Thank you again for your valuable feedback, which has greatly improved our work.
>
> Best regards,
>
> Authors

---

### Official Review · Reviewer_2Qhq · 2025-11-01

**Soundness:** 2
**Presentation:** 2
**Contribution:** 1
**Rating:** 4
**Confidence:** 2

**Summary:**

This paper introduces LearnAlign, an efficient data selection method for RL post-training, particularly in RLVR (Reinforcement Learning from Verifiable Rewards) settings. LearnAlign aims to improve data selection efficiency by incorporating a metric that is scaled by the success rate of each data point, in addition to considering the cosine similarity between the gradients of data.

**Strengths:**

- The authors propose a novel "Data Learnability" metric, which utilizes success rates to identify which data points possess more significant gradients. The experiments provided demonstrate that this metric can lead to more effective data selection.

**Weaknesses:**

- The novelty of LearnAlign appears limited when compared to LESS [1]. The core architecture seems largely similar, with the main difference being the use of verifiable rewards to compute a success rate, which in turn is used to scale the data importance metric.
- The empirical results in Tables 1-3 are presented solely as average performance. To properly evaluate the statistical significance of the claimed improvements, it is recommended that the authors also report standard deviations or confidence intervals across multiple runs.
- Based on the results in Table 3, the contributions of key components like "warmup" and "data learnability" do not appear to be significant. The "data learnability" metric is presented as a central component of the proposed method, yet its inclusion seems to yield minimal performance benefits, which raises questions about its practical impact.

[1] M. Xia, et al., “LESS: Selecting Influential Data for Targeted Instruction Tuning,” ICML 2024.

**Questions:**

Please refer to the points raised in the Weaknesses section above.

---

> ### Author Response · Authors · 2025-11-25
>
> Thank you very much for your comments on this paper. We address your concerns as follows:
>
> > Q1: The novelty of LearnAlign appears limited when compared to LESS.
>
> **A1:** We argue that LearnAlign exhibits significant differences from LESS in the following aspects:
>
> **1. Emphasis on Data Learnability for RLVR post-training**: While LESS primarily relies on gradient direction information, LearnAlign incorporates data learnability—accounting for difficulty and learning potential—in addition to gradient signals. As highlighted in prior studies [1,2], effective RL post-training requires alignment between task difficulty and the model’s capabilities. We introduce a learnability weight  $V(ξ)=p(1−p)$, which peaks at moderate difficulty and naturally emphasizes samples that the model can benefit from.  As for the theoretical motivation of data learnability, please refer to **A1** for Reviewer HRzu.
>
> **2. Novel Integration of Learnability and Gradient Information**: LearnAlign introduces a new approach to combine data learnability with gradient-based criteria. This framework also opens avenues for future research into developing more refined learnability metrics, e.g. considering the varying difficulty during the training process and the exploratory nature of the data.
>
> **3. Different Objectives**: LESS focuses on selecting instruction-tuning data tailored to specific capabilities reflected in a validation set, whereas LearnAlign is designed to select RLVR training data that efficiently enhances reasoning abilities.
>
> [1]  Proximal curriculum for reinforcement learning agents. arXiv 2023
>
> [2] How difficulty-aware staged reinforcement learning enhances LLMs' reasoning capabilities: a preliminary experimental study. arxiv 2025
>
> > Q2:Tables 1–3 report only average performance; standard deviations or confidence intervals across multiple runs are recommended.
>
> **A2:** Thanks a lot for your valuable suggestions. Following your suggestions, we  conduct the evaluation three times and reported both the average performance and the standard deviations. The results on DAPO-MATH-17K with Qwen2.5-3B model are shown in Table 1-1. More results will be updated later.
>
> Table 1-1.  Comparison of data selection methods on five benchmarks. We train Qwen2.5-3B on the DAPO-MATH-17K selected subset with 1,000 data points.
> | Data Selection Method | GSM8K | MATH500 | AMC2023 | AIME2024 | CURX |
> | --------------------- | ----- | ------- | ------- | -------- | ---- |
> | Random Sampling       | $73.4\pm0.4$      |   $57.3\pm0.9$      |   $24.7\pm0.9$    |      $12.2\pm1.5$      |    $17.4\pm 1.1$  |
> | PPL-Top               | $74.0\pm1.2$      |  $58.0\pm2.2$        |  $25.2\pm0.3$        |       $13.3\pm4.7$  |  $19.1\pm0.7$   |
> | PPL-Middle            | $72.2\pm0.2$      |  $52.7\pm0.5$       |   $23.5\pm 0.5$      |  $14.4\pm4.2$        |  $18.9\pm1.6$    |
> | IFD                   |   $69.6\pm0.4$    |  $56.0\pm1.8$       |   $22.7\pm0.4$      |  $15.5\pm6.8$        |  $17.9\pm 0.2$    |
> | TokenLength           |  $63.3\pm0.8$     | $52.7\pm1.3$        |    $20.4\pm0.1$     |     $13.3\pm5.4$     |   $18.1\pm0.2$   |
> | SelectIT              |  $68.8\pm0.7$     |   $55.1\pm0.9$      |   $23.1\pm1.6$      |    $13.3\pm0.0$      |   $14.7\pm0.4$  |
> | LIMR                  |  $73.6\pm0.8$    |   $57.3\pm1.5$      |   $23.3\pm0.4$  |      $15.6\pm3.1$   |   $17.1\pm0.3$ |
> | LearnAlign            |  $79.3\pm0.6$      |    $61.1\pm1.1$      |  $29.3\pm 1.4$      |   $16.7\pm3.2$       | $21.2\pm 0.9$     |
>
> > Q3: The significance of "warmup training" and "data learnability".
>
> **A3:** To fully show the role of "warmup training" and "data learnability", we conducted the ablation experiments on Qwen2.5-3B and Qwen2.5-7B  by training for more steps, and the results are shown in Table 1-2. As shown in Table 1-2, both of them have significant impact for the proposed method through sufficient training.
>
> Table 1-2. Ablation study of warmup training and data learnability. We train Qwen2.5-3B and Qwen2.5-7B  model on the DAPO-MATH-17K selected 1,000 examples.
>
> | **Benchmark**                                      | **GSM8K**       | **MATH500**     | **AMC2023**     |
> |----------------------------------------------------|------------------|------------------|------------------|
> | *LearnAlign (Qwen2.5-3B, 2,000 steps)*             | 83.8±1.0         | 67.8±2.1         | 36.9±0.7         |
> | **w/o warmup training (Qwen2.5-3B, 2,000 steps)**  | 81.9±0.2         | 64.4±0.6         | 31.0±2.0         |
> | **w/o data learnability (Qwen2.5-3B, 2,000 steps)**| 81.3±1.1         | 63.6±2.2         | 34.8±0.8         |
> | *LearnAlign (Qwen2.5-7B, 1,000 steps)*             | 90.4±0.4         | 76.7±0.4         | 48.6±0.3         |
> | **w/o warmup training (Qwen2.5-7B, 1,000 steps)**  | 89.9±0.1         | 73.2±0.2         | 43.7±0.3         |
> | **w/o data learnability (Qwen2.5-7B, 1,000 steps)**| 89.9±0.3         | 75.3±0.5         | 46.4±0.7         |

---

> ### Author Response · Authors · 2025-12-03
>
> Table 1-3 Comparison of data selection methods on GSM8K test set.We train Qwen2.5-1.5B-Instruct on the GSM8K training selected subset.
> Mean and standard deviation are reported.
> | **Data Selection Method**        | **100**        | **500**         | **1,000**        | **2,000**        |
> |----------------------------------|----------------|------------------|------------------|------------------|
> | Qwen2.5-1.5B-Instruct            | 55.7±0.8 |
> | Qwen2.5-1.5B-Instruct-FULL       | 77.0±0.3|
> | Random Sampling              | 74.0±0.7       | 74.8±0.5         | 74.9±0.1         | 75.8±0.2         |
> | PPL-Top                    | 72.2±0.3       | 73.1±0.1         | 73.8±0.6         | 75.3±0.4         |
> | PPL-Middle                   | 73.9±0.1       | 74.8±0.5         | 74.8±0.4         | 75.3±0.6         |
> | IFD                          | 74.1±0.7       | 76.0±0.3         | 75.5±0.4         | 76.1±0.5         |
> | Token Length                 | 74.0±0.3       | 75.4±0.5         | 75.3±0.4         | 76.3±0.7         |
> | SelectIT                     | 74.2±0.3       | 75.0±0.2         | 75.4±0.6         | 75.0±0.4         |
> | LIMR                         | 74.2±0.3       | 75.6±0.7         | 75.5±0.6         | 76.3±0.3         |
> | **_LearnAlign_**                 | **74.5±0.3**   | **76.8±0.5**     | **77.5±0.4**     | **78.0±0.3**     |
>
> Table 1-4. Comparison of data selection methods on five benchmarks. We train Qwen2.5-7B on the DAPO-MATH-17K selected subset with 1,000 data points.
> | **Data Selection Method** | **GSM8K** | **MATH500** | **AMC2023** | **AIME2024** | **CRUX** |
> |---------------------------|-----------|-------------|-------------|--------------|----------|
> | Qwen2.5-7B                | 27.0±1.1  | 66.1±1.5    | 17.9±0.3    | 17.8±1.9     | 25.7±0.8 |
> | Qwen2.5-7B-FULL           | 89.8±0.1  | 73.9±0.5    | 49.2±0.7    | 32.2±1.6     | 52.4±1.1 |
> | Random Sampling           | 82.9±0.5  | 64.5±0.5    | 30.2±0.5    | 23.3±0.0     | 41.0±0.5 |
> | PPL-Top                   | 83.6±0.8  | 64.4±0.4    | 27.4±0.4    | 27.8±3.1     | 44.0±1.5 |
> | PPL-Middle                | 79.5±1.2  | 64.6±0.5    | 27.8±0.1    | 17.8±3.1     | 38.9±0.6 |
> | IFD                       | 82.8±1.1  | 63.1±0.4    | 27.6±1.0    | 20.0±0.0     | 35.3±0.4 |
> | Token Length              | 78.7±0.9  | 62.1±2.1    | 25.8±0.4    | 23.3±4.7     | 34.4±2.9 |
> | SelectIT                  | 84.6±0.2  | 64.3±1.5    | 27.7±0.1    | 18.9±3.1     | 40.8±0.8 |
> | LIMR                      | 82.7±1.3  | 65.7±1.6    | 28.0±0.4    | 25.6±3.1     | 39.4±0.9 |
> | **_LearnAlign_**          | **87.7±0.7** | **71.0±0.4** | **34.0±1.4** | **28.9±3.1** | **43.1±0.2** |

---

> ### Author Response · Authors · 2025-12-03
>
> **Dear Reviewer 2Qhq,**
>
> Thank you very much for your time and for providing such constructive and insightful comments. We value your feedback and have conducted additional experiments and revisions to address your concerns.
>
> In response to your specific questions, we have made the following updates:
>
> **Clarification of Novelty:** We have further clarified the contributions of LearnAlign relative to LESS, emphasizing the role of data learnability for RLVR post-training, the new integration of learnability with gradient information, and the difference in overall objectives.
>
> **Statistical Robustness:** Following your suggestion, we conducted three-run evaluations and now report both average performance and standard deviations. The updated results for DAPO-MATH-17K with Qwen2.5-3B are presented in Table 1-1, 1-3, and1-4.
>
> **Ablation Studies:** To better demonstrate the significance of warmup training and data learnability, we conducted extended long-time training experiments. The results in Table 1-2 show their substantial contributions to performance.
>
> We have incorporated these analyses and results into the revised paper. We trust that these additional experiments and clarifications have fully addressed your concerns regarding novelty and robustness. Thank you again for your thoughtful feedback and for helping us improve our work.
>
> Best regards,
>
> Authors

---

### Author Response · Authors · 2025-11-21

We would like to thank all reviewers for their time and effort in providing valuable feedback. We will upload our responses to each question or comment as soon as possible. Due to limitations in computing resources, it still needs to take some time. We sincerely appreciate your patience and understanding.

---

### Author Response · Authors · 2025-12-03
**Summary of Our Responses during the Rebuttal Period**

**Dear Senior Area Chair and Area Chair,**

Thank you for handling our submission! Considering the heavy workload during this period, we would like to provide a concise summary of our reviews and the rebuttal process. We hope this assists you in efficiently navigating the comments and our responses.

**Restatement of Contributions:**
This paper addresses the critical bottleneck of data inefficiency in Reinforcement Learning from Verifiable Rewards (RLVR) for LLMs. In the following, our main contributions are summarized:

1. In this paper, we explore efficient data selection for RLVR post-training from the perspective of gradient alignment, a direction that has received limited attention in prior work.
2. We introduce **LearnAlign**, a novel data selection framework that constructs learnability-weighted gradient representations to measure influence between data points, where the learnability metric captures learning potential and addresses the response-length bias for gradient norms.
3. Comprehensive comparison with prior methods across five benchmarks and three LLMs clearly reveals the shortcomings of traditional SFT data selection methods and demonstrates that **LearnAlign** identifies high-value subsets (e.g., 1,000 samples) that match or exceed full-dataset performance.

**Our paper is supported by positive feedback across several dimensions:**
* **Significance & Motivation:** The paper tackles a practical and important bottleneck, the high cost and data inefficiency of RLVR (Reviewers **HRzu**), with  certain theoretical significance and insights from using gradient information (Reviewers **bKxL**).
* **Clarity:** The paper is well-written, easy to follow (Reviewers **JWPt**, **HRzu**), and the data selection process defined in the paper is quite reasonable and logically structured (Reviewer **HRzu**).
* **Effectiveness:** The method proposes a novel "Data Learnability" metric, leading to more effective data selection (Reviewers **2Qhq**), successfully identifying valuable subsets, and achieving competitive performance with significantly reduced data size (Reviewers **JWPt**, **HRzu**, **bKxL**).
* **Comprehensive Experiments:** The experimental section is comprehensive, covering baselines, ablations, and cost analysis (Reviewer **JWPt**), providing insights of how traditional SFT data selection methods perform in RLVR (Reviewer **bKxL**).

**We have actively addressed key concerns and misunderstandings through extensive additional experiments:**

* **Convergence & Performance Ceiling (Reviewer bKxL):** The reviewer was concerned that the subset might converge slower or hit a lower performance ceiling. In the rebuttal, we provided **step-wise convergence curves** and **extended training experiments (up to 2000 steps)**. Results show LearnAlign converges 31% faster than full training and eventually surpasses the full-dataset performance, proving there is no performance ceiling imposed by the selection.
 * **Statistical Significance and Robustness to the Warmup Randomness (Reviewer 2Qhq，HRzu):** We provided **standard deviations** across multiple runs (Table 1-1, 1-3, 1-4) and ablations (Table 1-2) to prove the statistical significance of our improvements. We confirmed via experiments (Table 3-1) that the method is robust to the randomness of warmup data.
* **Baseline Supplementation (Reviewer bKxL):** We implemented the suggested **Pass@N Score filtering** baseline. New results (Table 4-2) demonstrate that LearnAlign consistently outperforms this common heuristic, highlighting the necessity of gradient alignment.
* **Time Cost and Computation Efficiency(Reviewer JWPt, bKxL)**: We added a detailed time-cost comparison (Table 3-3) and report that the LearnAlign scoring step is CUDA-optimized, taking only several seconds. The added appendix explains efficient gradient estimation techniques, confirming that the method is computationally practical and scalable.
* **Novelty (Reviewer 2Qhq) and Theoretical Justification (Reviewer JWPt, HRzu):** We clarified the distinct differences between LearnAlign and LESS (specifically the learnability metric for RLVR). And we provide a deeper theoretical justification for the p(1−p) learnability metric, showing its links to Bernoulli variance, Fisher information, and the magnitude of policy gradients.
* **Representativeness vs. Diversity (Reviewer HRzu):** To assess whether our method benefits from diversity, we tested K-means–augmented variants and found that feature-level diversity yields no meaningful gains and is sensitive to k (Table 3-2). This supports prioritizing representativeness, as shallow feature diversity does not correlate with reasoning quality well.

We believe the additional experiments and clarifications have resolved the reviewers' concerns. We hope our work can contribute to more efficient RL training paradigms for the community.

Thanks again for your time and effort!

Best regards,

Authors of Submission #419

---

### Note · Authors · 2026-01-06

I have read and agree with the venue's withdrawal policy on behalf of myself and my co-authors.